



# The influence of Amazonian anthropogenic emissions on new particle formation, aerosol, cloud and surface rain

Xuemei Wang[1,2], Kenneth S. Carslaw[1], Daniel P. Grosvenor[1,3], and Hamish Gordon[2]

[1]School of Earth and Environment, University of Leeds, LS2 9JT, Leeds, United Kingdom
[2]Department of Chemical Engineering and Center for Atmospheric Particle Studies, Carnegie Mellon University, PA 15213, United States
[3]Met Office Hadley Centre, Exeter, EX1 3PB, United Kingdom

**Correspondence:** Kenneth S. Carslaw (k.s.carslaw@leeds.ac.uk)

**Abstract.** Anthropogenic emissions have been shown to affect new particle formation, aerosol concentrations, and clouds. Such effects vary with region, environmental conditions and cloud types. In the wet season of Amazonia, anthropogenic emissions emitted from Manaus, Brazil can significantly increase the cloud condensation nuclei (CCN) concentrations compared to the background of mainly natural aerosols. However, the regional response of cloud and rain to anthropogenic emissions in Amazonia remains very uncertain. Here we aim to quantify how new particle formation, aerosol concentration, cloud and rain respond to changes in anthropogenic emissions in the Manaus region and to understand the underlying mechanisms. We ran the atmosphere-only configuration of the HadGEM3 climate model with a nested regional domain that covers most of the rainforest region (720 km by 1200 km with 3 km resolution) under several regional emission scenarios. The 7-day simulations show that, in the areas that are affected by anthropogenic emissions, when aerosol and precursor gas emissions are doubled from the baseline emission inventories, aerosol number concentrations increase by 13 %. The nucleation rate that involves sulfuric acid generally increases with pollution levels. However, near the source of the pollution, nucleation is suppressed due to high primary aerosol emission, resulting in smaller nucleation and Aitken mode aerosol number concentrations. We also found that doubling the anthropogenic emission can increase the cloud droplet number concentrations ($N_d$) by 9 %, but cloud water mass and rain mass do not change significantly. Even very strong reductions in aerosol number concentrations by a factor of 4, which is an unrealistic condition, cause only 4 % increase in rain over the domain. If we assume our simulation has fine enough grid resolution and an accurate representation of the relevant atmospheric processes, the simulated weak response of cloud and rain would imply that the Amazonian convective environment is non-linear and resilient to the small changes in $N_d$ that occur in response to localised anthropogenic aerosol emissions.

## 1 Introduction

Anthropogenic aerosols contribute a high fraction of uncertainty in radiative forcing of climate change by acting as cloud condensation nuclei (CCN) (Kaufman and Nakajima, 1993; Jones et al., 1994; Kaufman and Fraser, 1997; Kaufman et al., 1998; Pierce and Adams, 2007; Wang and Penner, 2009; Merikanto et al., 2009; Kazil et al., 2010; Yu et al., 2013; Dunne et al., 2016; Gordon et al., 2016, 2017; Gryspeerdt et al., 2019, 2021). Several modelling studies have shown that anthropogenic emissions





can affect aerosol concentrations and CCN (Manktelow et al., 2009; Laakso et al., 2013; Yu et al., 2013; Chang et al., 2017;
Shrivastava et al., 2019; Zhao et al., 2021). Changes in CCN concentration influence cloud properties such as cloud droplet
number concentration ($N_d$), which then causes cloud adjustments of liquid water path (LWP) and cloud fraction (Twomey,
1977; Albrecht, 1989; Kamae et al., 2015). Evidence for the effect of aerosol and pollution on cloud optical depth, cloud thick-
ness, $N_d$ and precipitation have been confirmed by observational studies (Kawamoto, 2006; Sporre et al., 2012; Gonçalves
et al., 2015; de Sá et al., 2018; Fan et al., 2018; Douglas and L'Ecuyer, 2021). However, it is hard to interpret and quantify the
influences of anthropogenic emissions on clouds, especially for convective clouds which involve complex relationship between
aerosol particles, thermodynamic processes and cloud microphysics. In this study, we aim to investigate the extent to which
the anthropogenic emissions affect aerosol concentration, cloud, rain, and the underlying mechanisms.

Despite the difficulties, many previous studies have investigated the relationship between aerosol and convective clouds. Cec-
chini et al. (2016) used the observations from GoAmazon2014/5 (Observations and Modeling of the Green Ocean Amazon
2014-2015) and showed that under polluted conditions, the warm-phase cloud droplet effective diameter had changes of 10 %-
% and $N_d$ was differ by a factor of 10 vertically compared to background conditions (Cecchini et al., 2016). An increased
loading of aerosol particles can also influence the mass of liquid that condenses and/or freezes, releasing extra latent heat.
Hence, the change in cloud microphysics has the potential to affect cloud dynamics (e.g., updraft velocity), cloud fraction etc.
(Kawamoto, 2006; Rosenfeld et al., 2008; Marinescu et al., 2021). The response of clouds to increasing aerosol concentrations
may depend on aerosol sizes. Fan et al. (2018) showed that extra particles, even if their diameters are smaller than 50 nm, from
pollution plumes could form additional cloud droplets and release extra latent heat which would subsequently 'invigorate'
deep convection in Amazonia. This process refers to the strengthening of convective updrafts (Andreae et al., 2004; Rosenfeld
et al., 2008), and is a topic of much discussion (e.g. Lebo et al., 2012; Grabowski and Morrison, 2020, 2021; Igel and van den
Heever, 2021; Varble et al., 2023). Koren et al. (2010) used satellite data from MODIS and found that more aerosols could
cause taller clouds and larger anvils. A greater concentration of aerosol could also cause a higher cloud fraction (Koren et al.,
2005, 2008, 2010) and cloud top height (Koren et al., 2012). Zaveri et al. (2022) found that the rapid growth of particles at
a few nanometres in diameter could lead to the suppression of precipitation from shallow clouds and then, trigger shallow to
deep cloud transition.

Increasing aerosol concentrations can produce also more smaller-size ice crystals. These extra ice crystals are formed by in-
creased concentrations of cloud droplets due to high supersaturation levels in deep convective clouds (Khain et al., 2012; Fan
et al., 2013; Herbert et al., 2015; Grabowski and Morrison, 2020). Such increase in ice crystals may affect graupel formation
(van den Heever and Cotton, 2007; Khain et al., 2011; Wellmann et al., 2018; Li et al., 2021) and can form a greater anvil (Fan
et al., 2010; Morrison and Grabowski, 2011; Fan et al., 2013; Yan et al., 2014; Chen et al., 2017, 2020).

The large number of complex interacting processes in deep convective clouds (activation, autoconversion, accretion, sedimen-
tation, latent heat release etc.) implies that the effects of aerosol on precipitation in these clouds are likely buffered and vary
with region, background aerosols and environmental conditions (Khain and Pokrovsky, 2004; Fan et al., 2007; Tao et al., 2007;



Khain et al., 2008; Lee et al., 2008; Yuan et al., 2008; Fan et al., 2009; Khain, 2009; Stevens and Feingold, 2009; Fan et al., 2013; Storer et al., 2010; Connolly et al., 2013; Liu et al., 2016; Heikenfeld et al., 2019; Dagan et al., 2022). The influence of aerosol on deep convective systems is also buffered by large-scale meteorology (Morrison, 2012; Grabowski, 2018; Dagan

et al., 2022). Nonetheless, studies have found both reduction of light rain in some clouds, and enhancements of warm rain in others, due to increased aerosol concentrations (Qian et al., 2009; Li et al., 2011; Wang et al., 2011; Fan et al., 2012, 2013; Tao et al., 2012; Koren et al., 2012; Guo et al., 2014; Jiang et al., 2016). A continuous supply of CCN was found necessary to sustain storm clouds and extra sub-micron aerosol activation was found to invigorate deep convective clouds (Ekman et al., 2004; Fan et al., 2018), while adding large particles to the environment can cause a reduction of rain in mixed-phase clouds

(Pan et al., 2022). The suppression of ice clouds is because large CCN can directly activate and form warm rain (Feingold et al., 1999; Yin et al., 2000; van den Heever et al., 2006). As a result of this complexity, the effects of anthropogenic emissions on clouds via NPF and aerosols are still not well understood.

Amazonia is one of the most pristine environments in the present-day, especially during the wet season when rain cleans the air, but the environment is still affected by pollution from cities like Manaus in central Amazonia. Aircraft measurements over

Manaus and the downwind forest have shown that around 20 % of the total particulate matter at 1 $\mu$m diameter are composed of anthropogenic sources which include sulfates, nitrates and ammonium (Shilling et al., 2018). Observations from a research tower downwind of Manaus showed that the total sub-micron particulate matter concentration is up to a factor of 2 higher in polluted conditions than in background conditions (de Sá et al., 2018). Cirino et al. (2018) used observations from two towers downwind of Manaus to show that the fractional contribution of organic gas molecules to aerosol mass increased when the

sites were further away from emission sources, implying the decreasing influences of pollution with longer distance from the emission source. Glicker et al. (2019) reported higher particle concentrations during high-pollution days from observations and their back-trajectory model showed that the high concentrations were due to emissions from Manaus. Other modelling studies have also confirmed that anthropogenic emissions enhanced aerosol mass by a up to factor of 4 and enhanced number concentrations by a factor of 5-25 downwind of Manaus (Shrivastava et al., 2019; Zhao et al., 2021).

To study the effects of anthropogenic emissions on aerosol and cloud over Amazonia, especially for deep convective clouds, we use a regional model with high-resolution emissions and resolved convection nested inside a global model. We aim to answer the following two questions:

(1) What are the effects of anthropogenic emissions on aerosol, cloud and rain in Amazonia?

(2) What are the mechanisms that drive changes in aerosol and cloud properties?

Our paper is organised as follows. Section 2 presents the observations and model configurations as well as simulation details used in this study. The results are shown in Sect. 3. Section 3.1 shows the comparison between the regional model results and observations. Section 3.2 and 3.3 describe the effects of anthropogenic emissions on aerosol particles, cloud and rain profiles. We discuss and conclude the results in Sect. 4.



## 2 Methods

### 2.1 GoAmazon2014/5 campaign and G-1 aircraft observations

The observations used in this study are from the 2-year field campaign Observations and Modeling of the Green Ocean Amazon 2014-2015 (GoAmazon2014/5) in central Amazonia (Martin et al., 2016, 2017). The campaign aimed to study the response of the Amazonian environment under pollution plumes transported from Manaus in 2014 and 2015. The campaign included aircraft measurements onboard a low-altitude G-159 Gulfstream I (G-1) in February, March, August, September and October 2014. There were 9 fixed research sites that collected observations in various environments such as urban, forest, and pasture, both upwind and downwind of Manaus in the form of transects of the pollution plume and the surrounding areas. The measured data include meteorology, aerosol, gas pollutants, and cloud properties (Martin et al., 2016, 2017).

We used the aircraft measurements of aerosol number concentrations onboard the G-1 aircraft with a time interval of 1 second on 11, 12, 14, 16 and 17 March 2014. There were 15 flights available in February and March 2014. The selected five days are within our regional model simulation time (11-18 March 2014). Figure 1 shows the flight tracks of the selected five days, which are mainly transects of the plume from Manaus. The measured aerosol particles with diameters greater than 3 nm ($N_{D>3nm}$), 10 nm ($N_{D>10nm}$) and 100 nm ($N_{D>100nm}$) are compared with the model. $N_{D>3nm}$ and $N_{D>10nm}$ were measured using a Condensation Particle Counter (CPC) with diameter ranges of 3 nm - 3 mm and 10 nm - 3 mm, respectively. $N_{D>100nm}$ was measured with a Passive Cavity Aerosol Spectrometer Probe (PCASP). Full details of the instruments can be found in Martin et al. (2017). During the 5 days, most of the measurements were made below 2 km altitude, and occasionally at 2-6 km altitude where particle number concentrations were between around 100 and 200 cm$^{-3}$, significantly lower than the concentrations with diameters greater than 3 nm (18000 cm$^{-3}$) in the lowest 2 km layer.

We also used the aerosol size distributions measured at the T3 research tower (3.2 °S, 60.6 °W) which is southwest (downwind) of Manaus (Martin et al., 2016). The size distributions were measured using the Ultra-High Sensitivity Aerosol Spectrometer (UHSAS) for particles with diameters of 55-1000 nm. 3-hourly precipitation rates measured by the S-band Amazon Protection National System radar between 11-17 March 2014 are additionally used to evaluate the model.





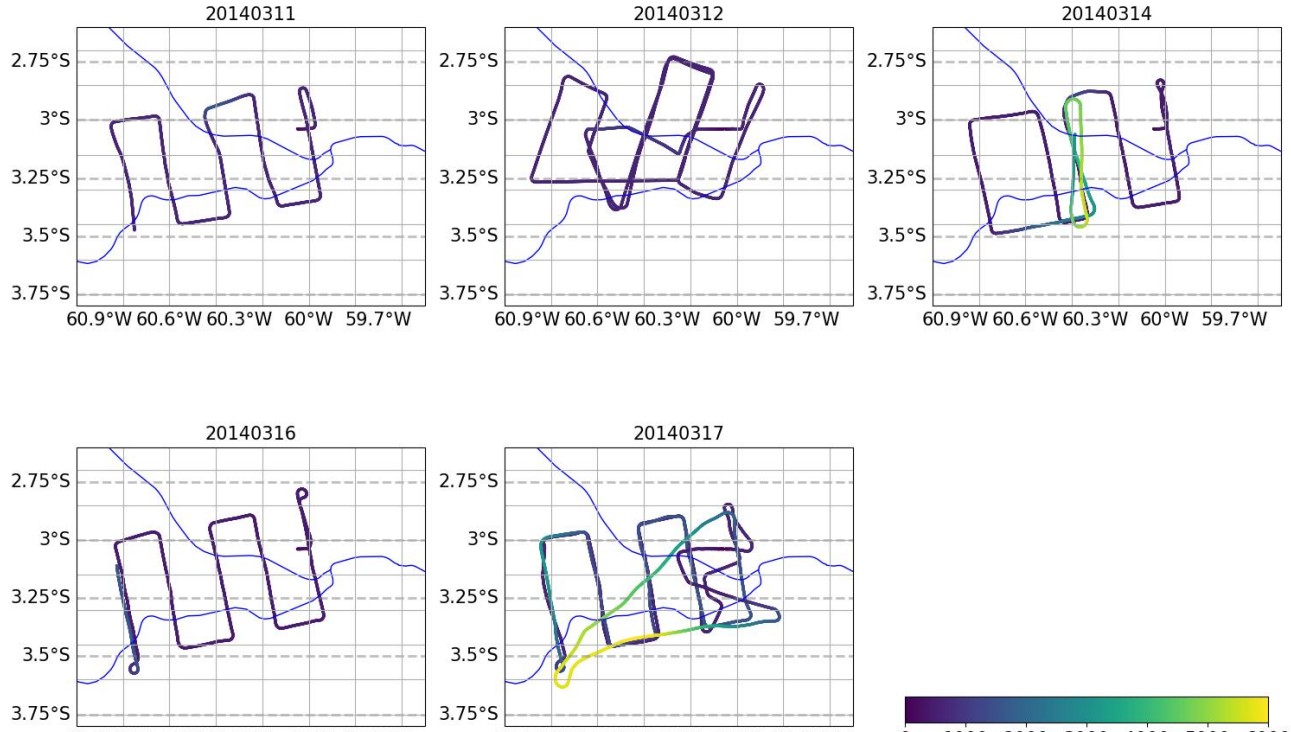

**Figure 1.** G-1 flight tracks on 11, 12, 14, 16 and 17 March 2014. The aircraft flew at below 2 km altitude on 11, 12 and 16 March and reached around 6 km on 14 and 17 March 2014. The colorbar indicates the flight altitude in m. The blue lines indicate the Rio Negro and the Amazon River.

## 2.2 Global and regional model configurations

We used a nested regional model located in central Amazonia, embedded in a global model. The global model is the atmosphere-only configuration of the Hadley Centre Global Environment Model version 3 (HadGEM3). Both models are based on the Uni-

fied Model (UM version 11.6), and both models are coupled to the UKCA (United Kingdom Chemistry and Aerosol) model (Planche et al., 2017; Gordon et al., 2018, 2023). The global and regional model are coupled in a one-way manner that allows the global model to drive the regional model with information including aerosols, trace gases, and meteorology conditions (temperature, 3D wind, cloud liquid, cloud ice, humidity and rain), while the global model is not affected by the regional model.

The global model uses the GA7.1 (Global Atmosphere v7.1) configuration of the UM with the Even Newer Dynamics for General atmospheric modelling of the environment (ENDGame) dynamical core (Wood et al., 2014; Walters et al., 2019). The



resolution is N96 (around 135 km) in the horizontal direction and there are 85 vertical levels up to 80 km in altitude. Parameterised convection is used in the global model (Fritsch and Chappell, 1980; Gregory and Rowntree, 1990; Stratton et al., 2009; Derbyshire et al., 2011; Walters et al., 2019).

The nested regional model is driven by the boundary conditions from the global model and its domain is centred at (3.1°S, 62.7°W), downwind of Manaus. The domain is 1200 km (east to west direction) by 700 km (north to south direction) with 3-km horizontal resolution. There are 70 vertical model levels with the highest altitude at 40 km. The lowest 64 levels extend from the surface to 20 km in altitude, which is the main region of interest for aerosol-cloud interactions. The regional model uses explicit convection which allows heat transfer and tracer transport to be resolved on the model grid.

The aerosol-chemistry scheme (UKCA) uses the GLOMAP-mode (Global Model of Aerosol Processes) two-moment aerosol microphysics model which allows aerosol to form from gaseous precursors, grow to larger sizes, and be transported and removed (Mann et al., 2010). The aerosol particles are represented by four water-soluble modes (nucleation, Aitken, accumulation and coarse) and an insoluble Aitken mode, which are specified by the number and mass (or equivalently size) with a fixed-width log-normal distribution. The particle chemical composition includes sulfate, sea-salt, black carbon and organic

carbon. Aerosol particles are scavenged by impaction with precipitation (washout) below clouds and are scavenged during the rain formation process (rainout), i.e. collision and coalescence of aerosol-containing cloud droplets with some of the aerosol assumed to be deposited to the surface. The removal processes of aerosols are size-dependent, and are controlled by a collection efficiency look-up table (Mann et al., 2010; Kipling et al., 2013).

UKCA uses a coupled chemistry scheme (StratTrop) which involves 84 species with 81 of them having chemical reactions
(Archibald et al., 2020), including several chemical reactions with anthropogenic gas species (ammonia, ethane, nitrogen monoxide etc.). StratTrop chemistry scheme can well represent reactions associated with pollution plumes from Manaus and the biogenic emissions from the surrounding forest in Amazonia, and subsequently affects NPF in this study. Here, monoterpenes are oxidised by the oxidants (OH, $O_3$ and $NO_3$), and the oxidants in StratTrop chemistry scheme can be transported, deposited as tracers, and vary the concentrations with chemical reactions. The StratTrop scheme has been used in global mod-
elling studies (Mulcahy et al., 2020), and was firstly incorporated in regional modelling in the study of Gordon et al. (2023).

Most of the emissions of anthropogenic gases and aerosols are obtained from the high-resolution (0.1°by 0.1°) EDGAR (Emissions Database for Global Atmospheric Research) inventories (Janssens-Maenhout et al., 2015). The fine grid resolution of these emissions allows us to resolve the Manaus pollution plume in our model. The emissions we use in the model are monthly means for the year 2010 and Table 1 shows all the included species. A diurnal cycle is applied for NO, BC and OC to
simulate the time variation of traffic. CMIP6 emission inventories provide $CH_4$, and monoterpene, isoprene and natural $SO_2$ are from CMIP5 inventories. The marine source of DMS has been parameterised based on Lana et al. (2011) and the land source is from biomass burning (van der Werf et al., 2006; Lamarque et al., 2010; Granier et al., 2011; Diehl et al., 2012). The emitted $CH_4$ from biomass burning data have been generated by the JULES model (Mangeon et al., 2016). Monoterpene and





isoprene are obtained from monthly mean emission inventories that include vegetation (Guenther et al., 1995; Pacifico et al.,
2012). Natural $SO_2$ comes from volcanic eruptions (Stier et al., 2005). The primary aerosols are emitted into the UKCA model
at a fixed geometric mean diameter of 150 nm.

**Table 1.** Gaseous species and aerosol emissions that are anthropogenic

| Species names and primary aerosol emissions | | | |
| --- | --- | --- | --- |
| BC | OC | $SO_2$ | $NH_3$ |
| NOx | $CH_3CHO$ | $CH_3COCH_3$ | $CH_2O$ |
| CO | $C_3H_8$ | $C_2H_6$ | Biomass burning aerosol |

The UKCA aerosol-chemistry model is coupled to the CASIM (Cloud-AeroSol Interacting Microphysics) cloud microphysics
scheme in the regional domain of the model for both stratiform and resolved convective cloud. CASIM is a two-moment cloud
microphysics model with five types of hydrometeor (cloud droplets, rain, ice, snow and graupel;  Shipway and Hill, 2012;
Hill et al., 2015; Grosvenor et al., 2017; Field et al., 2023). Aerosol number concentration and concentrations of the chemical
species are used by CASIM to calculate a weighted mean hygroscopicity for cloud droplet nucleation (Gordon et al., 2020).
CASIM then activates aerosols based on the mean gridbox updraft velocity and the activated prognostic $N_d$ is advected with the
resolved wind fields (Grosvenor et al., 2017; Miltenberger et al., 2018a). A diagnostic maximum supersaturation is calculated
to activate aerosols within the parameterization of (Abdul-Razzak and Ghan, 2000). The prognostic $N_d$ is replaced by newly
activated droplets if the newly activated concentration exceeds the existing concentration. Scavenging rates of aerosols during
convection are calculated from precipitation rates derived from the autoconversion and accretion rates in CASIM (Miltenberger
et al., 2018a).

## 2.3   New particle formation

New particle formation (NPF) represents the conversion processes from gas vapour to particle phases. The NPF mechanisms
include a scheme that includes sulfuric acid and organic gas molecules  ($H_2SO_4$-Org,  Riccobono et al., 2014) and the pure
biogenic nucleation, which uses oxidised organic gas molecules (Kirkby et al., 2016). The ability of biogenic vapour to nucleate
depends on vapour volatility (Weber et al., 1995; Vehkamäki et al., 2002; Riccobono et al., 2014; Dunne et al., 2016; Kirkby
et al., 2016; Tröstl et al., 2016; Simon et al., 2020). NPF in the UKCA model produces aerosol particles up to 3 nm in diameter.

Our model overestimates the aerosol number concentrations in the free and upper troposphere (Ranjithkumar et al., 2021).
Therefore, in our simulations only, we switch off all new particle formation above 1 km altitude and below 100 m altitude,
so that our model has a better representation of the observed particle number concentrations (Andreae et al., 2018; Shilling
et al., 2018). After switching off NPF at these altitudes, the model produced much lower aerosol concentrations than with NPF.
Although, we do find NPF at these altitudes to be the dominant source of aerosols smaller than 100 nm in diameter (Fig. A1,
A4 and A5) as expected, we parameterise the NPF processes as follows.





Based on the CLOUD chamber experiments, we use the inorganic-organic ($H_2SO_4$-Org) combined nucleation mechanism which has been parameterised in GLOMAP (Riccobono et al., 2014). New particles are formed via gas-to-particle conversion and condensation of stable highly-oxygenated molecules (Ehn et al., 2014; Kirkby et al., 2016; Tröstl et al., 2016; Stolzenburg et al., 2018; Bianchi et al., 2019) and can continue to grow through the condensation of HOM1 and $H_2SO_4$. The HOM1 is an oxidation product of monoterpenes, oxidised by OH (Gordon et al., 2016). Here, monoterpene is a type of BVOC (Biogenic

Volatile Organic Compound) emitted from forests by the UKCA model. Nucleation rates at 1.7 nm diameter are derived using the concentrations of HOM1 and $H_2SO_4$.

$$J_{H_2SO_4-Org_{1.7nm}} = \exp(-(T-278)/10) \times (0.5 \times k \times [H_2SO_4]^2 \times [HOM1]), \tag{1}$$

where, $J_{H_2SO_4-Org_{1.7nm}}$ represents nucleation rate at 1.7 nm in $cm^{-3}$ $s^{-1}$, [HOM1] and [$H_2SO_4$] represent the concentrations in molecules per $cm^{-3}$, k is kinetic factor with a constant value ($3.27 \times 10^{-21}$ $cm^6$ $s^{-1}$) (Riccobono et al., 2014). The

nucleation rates are multiplied with a temperature dependency $\exp(-(T-278)/10)$ and therefore nucleation rates vary with altitude (Gordon et al., 2016; Simon et al., 2020).

We also include the pure biogenic nucleation mechanism following Kirkby et al. (2016); Gordon et al. (2016), but the biogenic nucleation is not expected to significantly influence the particle concentrations between 100 m and 1 km altitude compared to the $H_2SO_4$-Org mechanism. This NPF parameterisation produces particles at 1.7 nm diameter using HOM2. In the parame-

terisations, HOM2 is oxidised from monoterpenes by OH and $O_3$, and HOM2 concentrations are obtained by a steady-state approximation (Gordon et al., 2016). Yields of HOM2 are 1.2 % when monoterpenes are oxidised by OH and 2.9 % by $O_3$ and the concentrations of HOM2 are used to derive nucleation rates for particles at 1.7 nm in diameter following Gordon et al. (2016). The nucleation rate sums up the neutral and ion-induced nucleation rate. In this study, for simplicity, the ion-induced nucleation uses a constant ion concentration of 400 $cm^{-3}$ ([Ion] in eq. 2),

$$J_{Bio_{1.7nm}} = \exp(-(T-T_0)/10) \times (A_1 \times ([HOM2]/10^7)^{\frac{A_2+A_5}{[HOM2]/10^7}} + [Ion] \times A_3 \times ([HOM2]/10^7)^{\frac{A_4+A_5}{[HOM2]/10^7}}), \tag{2}$$

where $J_{Bio_{1.7nm}}$ is the nucleation rate at 1.7 nm in $cm^{-3}$ $s^{-1}$, T is temperature in K, $T_0$ is a constant temperature (278 K), HOM2 represents the concentrations of HOM2 in molecules per $cm^{-3}$, and $A_{1-5}$ are constant parameters (Gordon et al., 2016). The nucleation rates are also multiplied with a temperature dependency $\exp(-(T-T_0)/10)$.

A cloud condensation sink term is additionally added to UKCA to suppress nucleation rates in cloudy regions (Kazil et al.,

2011; Wang et al., 2023). Commonly, a condensation sink that allows gases to condense onto existing aerosol particle surfaces instead of nucleating new particles is used. The addition of a cloud condensation sink enables gases to also condense onto cloud hydrometeor surfaces. It is obtained by assuming constant values for cloud droplet and ice crystal number concentrations (both at 100 $cm^{-3}$), which are used along with cloud liquid and ice water content to derive the radii of hydrometeors. We then obtain the condensation sink using Fuchs and Sutugin (1971):




$$\mathrm{CCS} = 4\pi\mathrm{D_v} \times \mathrm{N_{hyd}} \times (\mathrm{r_{cloud}} + \mathrm{r_{ice}}), \tag{3}$$

where, CCS denotes cloud condensation sink in s$^{-1}$, $\mathrm{D_v}$ is the gas diffusion coefficient, $\mathrm{N_{hyd}}$ is a constant concentration of cloud hydrometeors (droplets or ice; 100 cm$^{-3}$), $\mathrm{r_{cloud}}$ and $\mathrm{r_{ice}}$ are the radii of cloud droplets and ice, respectively.

### 2.4  Simulation details

The global and regional models were run from 11 to 18 March 2014, covering the five research flights during GoAmazon2014/5 in the Amazonian wet season (Martin et al., 2016, 2017). The global model was run 69 days prior to the start of the regional simulation for the initialisation of the aerosol fields.

**Table 2.** Summary of model simulations detailing the different anthropogenic emissions and nucleation mechanisms used.

|  | Gas emission | Primary aerosol emission | Biogenic nucleation | $H_2SO_4$-Org nucleation |
|---|---|---|---|---|
| CTL | ✓ | ✓ | ✓ | ✓ |
| offREG |  |  | ✓ | ✓* |
| 0.5×emis | ✓0.5× | ✓0.5× | ✓ | ✓ |
| 1.5×emis | ✓1.5× | ✓1.5× | ✓ | ✓ |
| 2×emis | ✓2× | ✓2× | ✓ | ✓ |
| 5×emis | ✓5× | ✓5× | ✓ | ✓ |
| Prim_emis |  | ✓ | ✓ |  |
| 0.25×aero | ✓ | ✓ | ✓ | ✓ |
| 4×aero | ✓ | ✓ | ✓ | ✓ |
| CTL 1-month | ✓ | ✓ | ✓ | ✓ |
| offREG 1-month |  |  | ✓ | ✓* |
| CTL+Bn | ✓ | ✓ | ✓ | ✓ |

Note: The $H_2SO_4$-Org nucleation relies on $H_2SO_4$, an anthropogenic gas precursor emitted in the regional model and advected from the global model through the model boundaries. When anthropogenic emissions in the regional domain are set to zero in the 7-day and 1-month simulations (offREG and offREG 1-month), $H_2SO_4$-Org nucleation will still occur due to the small amount of $H_2SO_4$ advected from the global model. However, the height- and domain-mean $H_2SO_4$-Org nucleation rate at 100 m - 1 km in the regional model is reduced by a factor of 3000 after removing all anthropogenic emissions. Consequently, even though $H_2SO_4$-Org nucleation is included in these 2 simulations, the resulting nucleation rates are too small to produce a signaficant number of aerosols.





Table 2 summarises the simulations. All the simulations use NPF between 100 m and 1 km and include the cloud condensation sink.

The control (CTL) emission simulation includes both anthropogenic gas and primary aerosol emissions, and the offREG (off regional) simulation has anthropogenic emissions switched off in the regional domain. The species that are switched off in offREG (see Table 1) include anthropogenic gas emissions and primary aerosol emissions, as well as NO, NVOC from anthropogenic sources, BC and OC. Because the $H_2SO_4$-Org nucleation mechanism is strongly controlled by the concentrations of $H_2SO_4$ and advection from the global model cannot supply enough $H_2SO_4$ below 1 km to this region for nucleation, switching off emissions in the regional domain almost disables this nucleation process. We perturb all anthropogenic emissions by factors of 0.5, 1.5, 2 and 5 in additional simulations to understand the sensitivity of aerosols and cloud properties. The effects of primary anthropogenic aerosol emissions can be determined from the Prim_emis (primary emission) simulation where only anthropogenic primary aerosol emissions are kept and the $H_2SO_4$-Org nucleation is switched off in the regional domain to prevent secondary aerosol formation from anthropogenic gas precursors ($H_2SO_4$). The primary aerosol contribution to the total particle concentration and cloud properties can be derived with the equation $100\% \times (\mathrm{Prim\_emis} - \mathrm{offREG})/\mathrm{CTL}$. Two extra simulations were performed in which the the aerosol concentrations passed from UKCA to the CASIM aerosol activation process were scaled down by a factor 4 (simulation $0.25 \times$ aero) and scaled up by a factor of 4 (simulation $4 \times$ aero). The purpose of these simulations was to force a direct change in cloud droplet numbers compared to the perturbations achieved by changing emissions. As shown below, the 7-day simulations of the six emission scenarios showed an insignificant response of cloud properties to reductions in aerosol emissions, therefore the CTL and offREG simulations were also run for a month so that a longer-term effect on the clouds could be quantified.

## 3 Results

### 3.1 Comparison with observations

Figure 2 shows the timeseries of the observed and simulated particle number concentrations with diameters greater than 3 nm ($N_{D>3nm}$), 10 nm ($N_{D>10nm}$) and 100 nm ($N_{D>100nm}$) over the 5 days with aircraft observations from 11 to 17 March 2014. As shown in Fig. 1, the G-1 aircraft measured particle number concentrations in Manaus pollution plumes and for most of the time flew downwind of the city. Therefore, several peaks in particle number concentrations were observed during the flights. The modelled results in the CTL and offREG simulations are interpolated according to the flight time, coordinates and altitude for comparison with the observations.

All the observed particle concentrations ($N_{D>3nm}$, $N_{D>10nm}$ and $N_{D>100nm}$) exhibit strong temporal as well as spatial variations that are related to pollution plumes from Manaus. Among the five days of measurement, 16 and 17 March have the greatest number concentrations for all particle size ranges (around 11000 cm$^{-3}$ for $N_{D>3nm}$, 3200 cm$^{-3}$ for $N_{D>10nm}$, 270 cm$^{-3}$ for $N_{D>100nm}$ averaged over time) which implies that the downwind air was most polluted on 16 and 17 March and the





plumes were most distinct from the surrounding environments. The background number concentrations are around 1000 cm$^{-3}$
for $N_{D>3nm}$ and $N_{D>10nm}$, and around 300 cm$^{-3}$ for $N_{D>100nm}$ during the five days. The least polluted day is 12 March when
250   the time-mean particle number concentrations are 1300 cm$^{-3}$ ($N_{D>3nm}$), 900 cm$^{-3}$ ($N_{D>10nm}$), and 75 cm$^{-3}$ ($N_{D>100nm}$),
and the variability of $N_{D>3nm}$ is about 8 times smaller than the time-mean $N_{D>3nm}$ on 16 and 17 March. On the other two
days (11 and 14 March), the time-mean particle number concentrations are factors of around 1.6-3 smaller than concentrations
on 16 and 17 March for $N_{D>3nm}$ and $N_{D>10nm}$, and factors of 0.6-1.1 for $N_{D>100nm}$.

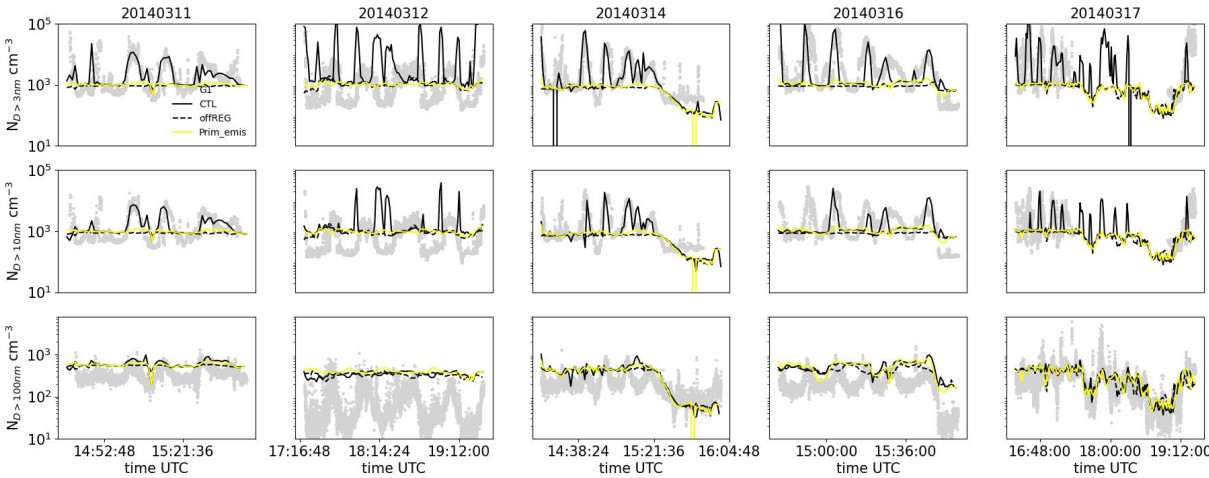

**Figure 2.** Timeseries of observed (grey dots) and simulated (CTL - black solid, offREG - black dashed and Prim_emis - yellow; solid lines)
particle number concentrations with diameters greater than 3 nm (upper row), 10 nm (middle row) and 100 nm (lower row) on 11, 12, 14, 16
and 17 March 2014. The observations were measured onboard the G-1 aircraft during the GoAmazon2014/5 campaign and model data are
interpolated according to the G-1 flight tracks.

The CTL simulation reproduces most of the observed in-plume number concentrations for $N_{D>3nm}$ and $N_{D>10nm}$, and most
255   of the temporal evolution for $N_{D>100nm}$. The modelled particle concentrations of the three size ranges well reproduce the
observations on 11, 14, 16, and 17 March 2014, but the particle concentrations are overestimated on 12 March 2014. Of all
the five days, the simulations are the closest to the observations on 11 March, with an mean bias of -8 % for $N_{D>3nm}$; for
$N_{D>10nm}$ the bias was around -3 %; the model overestimates $N_{D>100nm}$ by 70 %. On 14 and 16 March, particle number
concentrations are generally overestimated by the model by between 15 % and 20 % for $N_{D>3nm}$, underestimated by around
260   25 % to 28 % for $N_{D>10nm}$, and overestimated by between 63 % and 130 % for $N_{D>100nm}$. Overall, the model captures
the spatial and temporal variations well. On 17 March 2014, $N_{D>3nm}$ and $N_{D>10nm}$ are underestimated by around 20 % and
40 %, and $N_{D>100nm}$ is overestimated by 10 %. The comparisons are worse on 12 March for all three size ranges, with the
modelled particle concentrations being factors of 11 ($N_{D>3nm}$), 2 ($N_{D>10nm}$) and 3.6 ($N_{D>100nm}$) too high. This discrepancy
is related to large number of nucleation mode aerosol (Fig. A2). The burst of nucleation mode aerosols in the model is likely



caused either by the residuals of particles of all 3 size modes from 11 March that have not been scavenged within a day (by 12 March) because the background particle concentrations are around a factor of 3 higher than observed, or because the surface emissions in UKCA on 12 March are higher than reality. The CTL simulation produces similar magnitudes of precipitation compared to the precipitation measurement by the S-band radar during GoAmazon2014/5 (Fig. A3), but whether the modelled rain removed the same number of aerosol particles as in reality remains unknown. Overall, the modelled results reproduce most

of the observed temporal and spatial variations for $N_{D>100nm}$ during the five days, but with some overestimations.

We also compared the aerosol size distributions at the location of the T3 research tower (3.2 °S, 60.6 °W) for the CTL and CTL+Bn (binary nucleation $H_2SO_4$-$H_2O$) simulations (see Fig. A4 and A5). The results show that in the simulation with binary nucleation in the upper troposphere (CTL+Bn), the modelled aerosol size distributions are closer to the observations than without upper tropospheric nucleation in both the simulation and the model spinup period (CTL simulation), although CTL+Bn

does not perfectly reproduce the observations. The time series of the particle number concentrations (Fig. A1) show that the CTL+Bn simulation significantly overestimated the observed particle (diameters greater than 3 nm) number concentrations by factors of between around 2 and 44, producing too many particles. These results imply the importance of nucleation in the upper troposphere. The CTL simulation has more realistic particle concentrations than CTL+Bn and is able to reproduce the temporal and spatial evolution of the aircraft measurements. Thus, we use this simulation (CTL) as a baseline for our sensitivity

test to anthropogenic emissions.

## 3.2 Effects of anthropogenic emissions on aerosol

In this section, we investigate the influence of anthropogenic emissions along the G-1 aircraft flight tracks on $N_{D>3nm}$, $N_{D>10nm}$ and $N_{D>100nm}$ particles. We also evaluate the effects of emissions on aerosol and cloud profiles in the regional domain but only the areas that are affected by pollution, which we define according to the total sulfur species. High-sulfur

regions are defined according to the instantaneous column integrated sulfur content from both $H_2SO_4$ (sulfric acid) and $SO_2$ in the lowest 2 km, calculated as $\int_{z=0}^{z=2}(1000\rho_z S_z)\,dz$. Here z is altitude, $\rho_z$ is air density at a height of z, and $S_z$ denotes sulfur mass mixing ratio obtained from both $H_2SO_4$ and $SO_2$. A threshold value of $6\times10^{-5}$ g m$^{-2}$ is chosen to represent polluted conditions. These high-sulfur regions defined by the CTL simulation are used also for the other simulations (offREG, 0.5×emis, 1.5×emis, 2×emis 5×emis, 0.25×aero and 4×aero) for consistency, irrespective of the sulfur content in the other

simulations. Here, we only analyse the data to the west of the red line in the regional domain for each simulation (Fig. 3) because it represents the regions downwind of Manaus that are likely affected by Manaus pollution. The areas to the east of the red line are not included in the following analyses, but are needed as a part of the regional domain in order to allow space for air mass entering the regional domain at the eastern boundary to evolve before reaching the regions of interest. We understand that sulfur alone may not be able to mark all the regions that are affected by anthropogenic emissions in the domain, but it has the

closest relationship to NPF of all the emissions in our simulations. Figure 3 shows example definitions of where the high-sulfur values (within the contours) are at 21 UTC on 14 March 2014; most of the high-sulfur regions are around the Amazon river.




Although the high-sulfur regions evolve with time, Manaus, Tapauá and other riverside areas (where most of the cities are located) are always the most polluted regions in the regional domain.

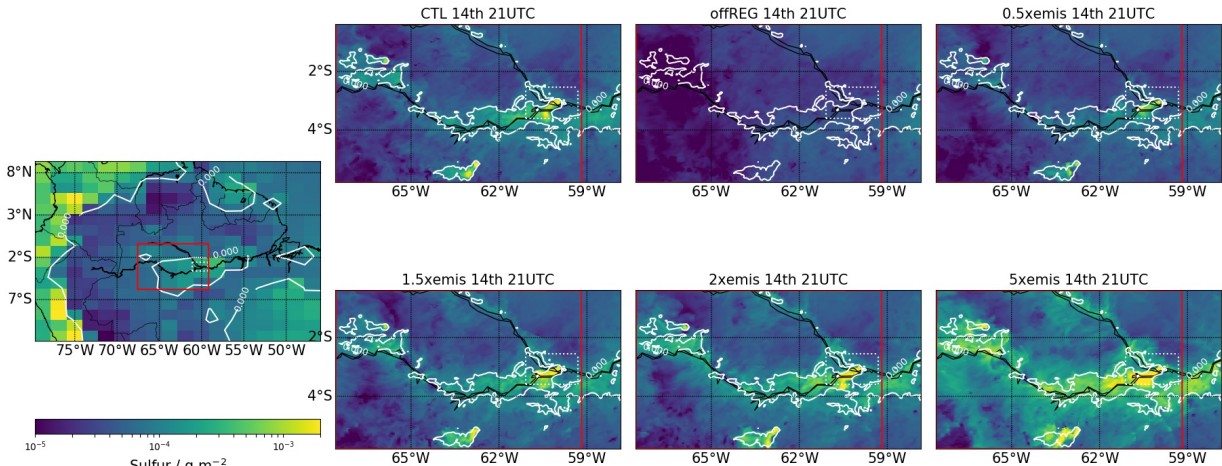

**Figure 3.** Maps of column integrated sulfur (g m$^{-2}$) at 21 UTC on 14 March 2014 in the CTL simulation in the global model (left), and in the CTL, offREG, 0.5×emis, 1.5×emis, 2×emis and 5×emis simulations in the regional model. The dotted rectangles mark where the G-1 aircraft flew in March 2014. The white contours in all the maps denote column integrated sulfur equal to $6 \times 10^{-5}$ g m$^{-2}$ in the CTL simulation. The area within the red box in the map of the global model and the area to the east of the red lines in the regional model mark the high-sulfur region.

Figure 2 also shows the particle number concentrations along the flight tracks when both anthropogenic gas and primary aerosol
emissions are switched off in the regional domain (offREG simulation) and when anthropogenic gas emission and H$_2$SO$_4$-Org nucleation are switched off in the regional domain (Prim_emis simulation). In the offREG simulation, the temporal and spatial variations of N$_{D>3nm}$ and N$_{D>10nm}$ are very small compared to the much larger variations in the CTL simulation. Although N$_{D>100nm}$ in the offREG simulations captures the background values, it misses most of the peak values in N$_{D>3nm}$ and N$_{D>10nm}$. The lack of temporal and spatial variability in the offREG simulation indicates that the variability shown in the
CTL simulation is caused by emission and NPF in the region, especially for N$_{D>3nm}$, which is reduced by 70-90 %, and N$_{D>10nm}$, which is reduced by 50-70 % during the 5 days compared to the CTL simulation. N$_{D>100nm}$ is least affected (6-20 % reduction) by anthropogenic emissions. Switching off anthropogenic emissions causes a reduction in the mean nucleation rates (biogenic and H$_2$SO$_4$-Org) along the track by up to a factor of $2.4 \times 10^5$ (16 March). On the same day, the condensation sink is reduced by a factor of 125 in the offREG simulation: the effect of anthropogenic emissions on nucleation is substantial.
N$_{D>100nm}$ has both increases and reductions in number concentrations when we switch off anthropogenic emissions with the reductions dominating for most of the time. The increases in N$_{D>100nm}$ at certain times may be caused by the suppression of NPF when there are no anthropogenic emissions, which thereby allows more condensable gases for particle growth (Sullivan et al., 2018), but we did not perform simulations that would allow the investigation of these changes. The occurrence of both increases and decreases in N$_{D>100nm}$ for CTL vs offREG implies that the effect of anthropogenic emissions in our simulations





on CCN is quite variable.

The temporal and spatial variations of $N_{D>3nm}$ and $N_{D>10nm}$ in the Prim_emis simulation are similar to those in the offREG simulation (Fig. 2). For most of the time, the Prim_emis simulation reproduces the observed $N_{D>100nm}$ while missing some peak concentrations. Compared to the offREG simulation, the Prim_emis simulation has a few more overlaps with the CTL for $N_{D>10nm}$ and $N_{D>100nm}$, indicating the contribution of large primary anthropogenic aerosol particles. The missing peak
concentrations in Prim_emis show that the discrepancies between the Prim_emis and CTL simulations are mainly caused by NPF induced by the anthropogenic emissions ($H_2SO_4$-Org mechanism). The contribution of primary aerosols to the region in which the G-1 aircraft flew is less than 3 % for $N_{D>3nm}$, between 1 % and 10 % for $N_{D>10nm}$, and less than 20 % for $N_{D>100nm}$. The contribution of primary aerosol to the total mean particle concentrations in the lowest 4 km of the atmosphere in the high-sulfur region of the regional domain is around 0.5 %. Thus, the majority of the changes in concentrations are caused
by the combination of precursor gas emission and NPF.

To better understand the response of aerosols and clouds to anthropogenic emissions using our model, we increased the anthropogenic gas and primary aerosol emissions (1.5×emis, 2×emis, and 5×emis simulations), which amplifies the differences in emissions between the offREG and CTL simulations. We also test the effect of reducing the emissions (0.5×emis). Figure 4. a, b and c shows the vertical profiles of aerosol number concentrations averaged over the high-sulfur regions. The concentrations
have similar shapes with height in all the six simulations. The number concentrations are the greatest below 2 km for all three modes of aerosol in the six simulations. In the CTL simulation the height-mean concentrations below 2 km are 130 cm$^{-3}$ for nucleation mode, 530 cm$^{-3}$ for Aitken mode, and 430 cm$^{-3}$ for accumulation mode. Because we prevented NPF above 1 km altitude, the particle concentrations are very low in the upper troposphere. Above 2 km, the aerosol number concentration quickly falls to very low concentrations at 6 km in altitude and remains at very low concentrations above 6 km in the CTL
simulation.





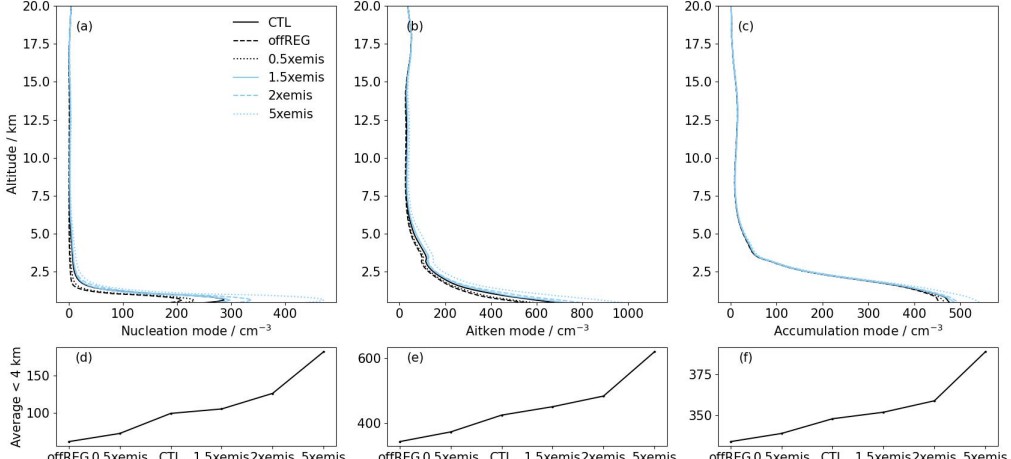

**Figure 4.** Profiles of (a) nucleation, (b) Aitken and (c) accumulation mode aerosol number concentrations, averaged over time and the area of the high-sulfur region (upper panel). Results are shown for the CTL (black solid), offREG (black dashed), 0.5×emis (black dotted), 1.5×emis (light blue solid), 2×emis (light blue dashed), and 5×emis (light blue dotted) simulations. The results are from the 3-hourly instantaneous model output. The lower panel is the nucleation mode (d), Aitken mode (e), and (f) accumulation mode aerosol concentration averaged over the lowest 4 km altitude of the profiles in the upper panel for the six simulations.

The influences of anthropogenic emissions on aerosol concentrations are quantified by the ratios of changes in aerosol concentrations to the factors of changes in anthropogenic emissions from the CTL simulation. Then, the means of the ratios are obtained, and later in the paper, we refer to the mean ratios as changes per unit of anthropogenic emissions, although the relationship may not be linear. The most significant changes in aerosol number concentration due to anthropogenic emissions among the six simulations exist in the lowest 4 km altitude. The height-mean nucleation mode aerosol number concentration below 4 km altitude changes by -38 to 82 cm$^{-3}$ in the five simulations with varied anthropogenic emissions compared to the CTL simulation (Fig. 4.d). On average, the nucleation mode aerosol concentration increases by 29 cm$^{-3}$ (29 % of the concentration in the CTL simulation) per unit increase in anthropogenic emissions. Similarly, Aitken mode changes by between -82 and 196 cm$^{-3}$ in each emission scenario and the concentration on average increases by 68 cm$^{-3}$ per unit increase in anthropogenic emissions (16 %; Fig. 4.e). The changes of accumulation mode range by -15 to 41 cm$^{-3}$ and on average the accumulation mode concentration increases by 12 cm$^{-3}$ (4 %) for each unit increase in anthropogenic emissions (Fig. 4.f). The total aerosol number, which also includes the insoluble Aitken mode and coarse mode, increases by around 113 cm$^{-3}$ (13 %) for each unit increase in anthropogenic emissions.





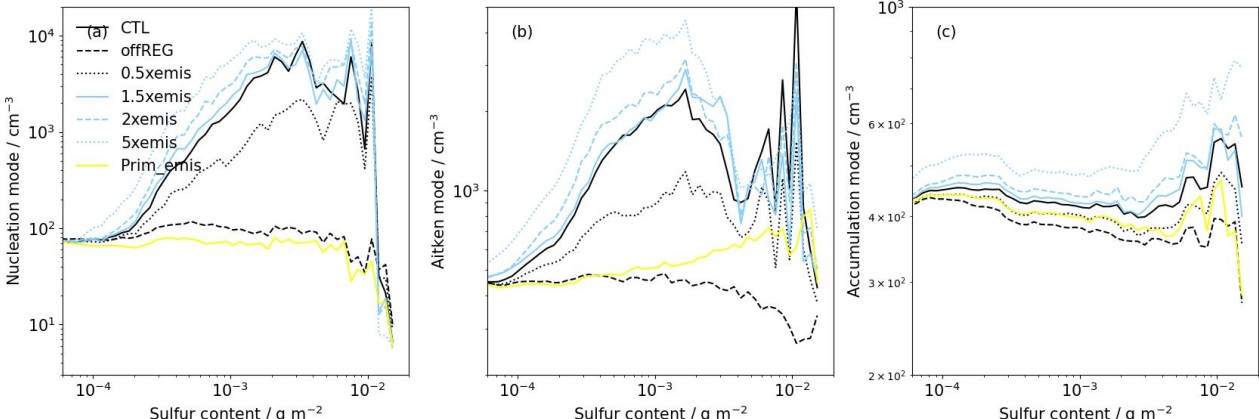

**Figure 5.** Dependence of (a) nucleation, (b) Aitken and (c) accumulation mode aerosol number concentrations in the lowest 2 km altitude on column integrated sulfur content. The concentrations are averaged over 100 sulfur content bins. Results are shown for the CTL (black solid), offREG (black dashed), 0.5×emis (black dotted), 1.5×emis (light blue solid), 2×emis (light blue dashed), 5×emis (light blue dotted), Prim_emis and (yellow) simulations. The results are from the 3-hourly instantaneous model output.

Figure 5 and 6 show the relationship between aerosol number concentration, nucleation rate, condensation sink, sulfuric acid and sulfur content in the lowest 2 km altitude, respectively. The lowest 2 km is used because this is where most of the pollution persists. The concentrations and rates are binned by sulfur content and each bin contains a mean. In all the simulations with anthropogenic gas emissions (CTL, 0.5×emis, 1.5×emis, 2×emis, and 5×emis), the concentrations of nucleation mode and Aitken mode aerosol increase with increasing sulfur content until around $2\text{-}3\times10^{-3}$ g m$^{-2}$. Subsequently, the concentrations significantly decrease in the largest sulfur content bin. Meanwhile, the accumulation mode aerosol number concentration remains relatively steady and starts to increase where the sulfur content is above $3\times10^{-3}$ g m$^{-2}$. The concentrations of accumulation mode aerosol are also reduced in the largest sulfur content bin, although the extent of reduction in concentration is much smaller than that of nucleation and Aitken mode aerosol.

The changes in nucleation mode aerosol concentration with sulfur content are closely related to the $H_2SO_4$-Org nucleation rate and sulfuric acid concentration, while Aitken and accumulation mode aerosols are less affected by nucleation rate. Overall, as anthropogenic emission in the regional domain increase, we find increases in aerosol particle concentrations for all size ranges, $H_2SO_4$-Org nucleation rate and condensation sink in each sulfur content bin. Although, the $H_2SO_4$-Org nucleation rate should be suppressed by a higher condensation sink as sulfur content increases, it is also enhanced by higher concentrations of sulfuric acid. This significant increase in the sulfuric acid concentration compensates for the suppression due to the condensation sink as sulfur content becomes larger.

The offREG simulation generally exhibits relatively small changes in aerosol concentrations, nucleation rate, condensation sink and sulfuric acid compared to other simulations with varied anthropogenic emission (CTL, 0.5×emis, 1.5×emis, 2×emis, and 5×emis) as sulfur content increases. The concentrations and rates in offREG simulation are usually several factors to orders





of magnitude smaller than those in the other five simulations, except for condensation sink. These low concentrations indicate the importance of anthropogenic emissions from a small region on particles through the $H_2SO_4$-Org nucleation process in our

model setup.

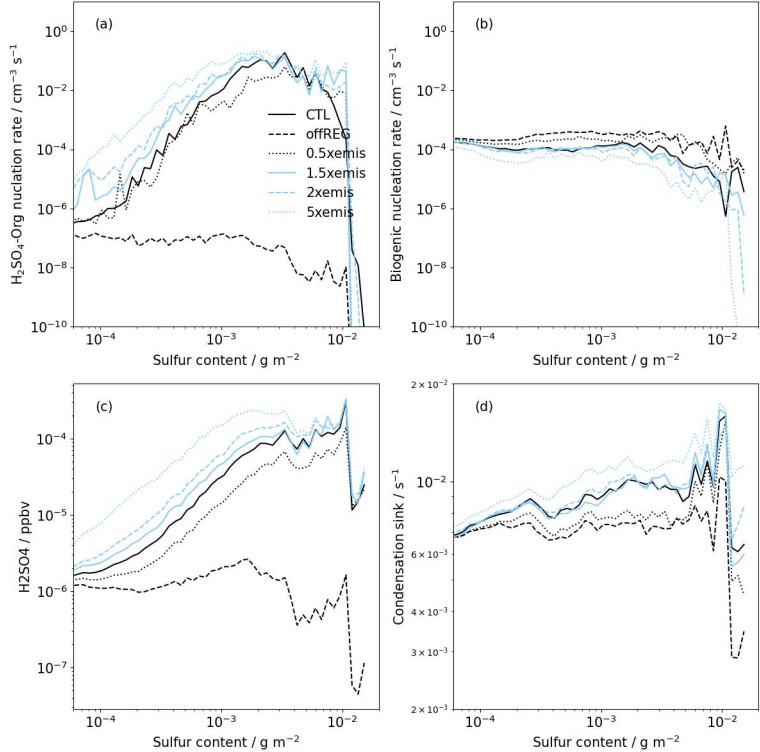

**Figure 6.** Correlations of (a) $H_2SO_4$-Org nucleation rate, (b) biogenic nucleation rate, (c) sulfuric acid concentration, and (d) condensation sink in the lowest 2 km altitude with column integrated sulfur content. The rates and concentrations are averaged over 100 sulfur content bins. Results are shown for the CTL (black solid), offREG (black dashed), 0.5×emis (black dotted), 1.5×emis (light blue solid), 2×emis (light blue dashed), 5×emis (light blue dotted) simulations. The results are from the 3-hourly instantaneous model output.

As sulfur content increases, the concentrations of nucleation and Aitken mode aerosol have a reduction of around a factor of 4 between 4 and $5 \times 10^{-3}$ g m$^{-2}$ of the sulfur content. This reduction is partly due to the rapidly increasing primary aerosol emissions in this sulfur content range (Fig. 5.b and c; yellow). In absence of anthropogenic gas emissions and $H_2SO_4$-Org nucleation, the Prim_emis simulation showed an increase in primary Aitken and accumulation mode aerosol number concen-

tration with increasing sulfur content between 4 and $5 \times 10^{-3}$ g m$^{-2}$. The amplified anthropogenic emissions in this sulfur content range can suppress nucleation and accelerate coagulation, resulting in more accumulation mode aerosols. The reduction of sulfuric acid by around a factor of 2 in this sulfur content range also implies that more aerosols act as a sink for sulfuric acid, which then suppresses aerosol nucleation.





In the largest sulfur content bin, the aerosol concentration of all sizes, nucleation rate, condensation sink, and sulfuric acid

have significant reductions. These data are collected very close to the pollution sources which are usually below 100 m where nucleation is not permitted. Additionally, emissions of primary aerosol near the source can also suppress sulfuric acid concentration and nucleation rate. The largest bin contains much less data and thus may not accurately represent the concentrations and rates.

### 3.3 Effects of anthropogenic emissions on cloud properties

Figure 7 shows the profiles of droplet number concentration ($N_d$) and ice number concentration ($N_i$) averaged over time and the cloudy areas in high-sulfur regions. The cloudy areas are defined as model grids with total cloud water content greater than 0.1 g kg$^{-1}$ and are defined separately for each simulation. In the CTL simulation, the mean $N_d$ in cloudy areas increases with height until around 1.3 km where it reaches a maximum of 135 cm$^{-3}$, then the concentration decreases until around 10 km altitude. $N_d$ profiles in the other simulations have similar shapes. Most of the differences that are caused by anthropogenic

emissions occur below 4 km and the relative magnitude follows the variations in aerosol concentrations in each simulation. The height-mean $N_d$ below 4 km altitude increase with increasing emissions, with values of 84 cm$^{-3}$ in offREG, 87 cm$^{-3}$ in 0.5×emis, 95 cm$^{-3}$ in CTL, 98 cm$^{-3}$ in 1.5×emis, 102 cm$^{-3}$ in 2×emis simulations, and 120 cm$^{-3}$ in 5×emis. While the 0.25×aero simulation has a mean $N_d$ of 36 cm$^{-3}$, which is a factor of approximately 0.38 of the CTL simulation and the 4×aero simulation has a mean $N_d$ of 224 cm$^{-3}$ (a factor of 2.4 of the CTL). Concluding from all the six simulations with

varied emissions, the height-mean $N_d$ in the lowest 4 km over time and cloudy areas in high-sulfur regions increases by 9 cm$^{-3}$ for each unit increase in anthropogenic emissions (equivalent to 9 % of the CTL simulation), but the latter two simulations produce more significant changes in $N_d$.





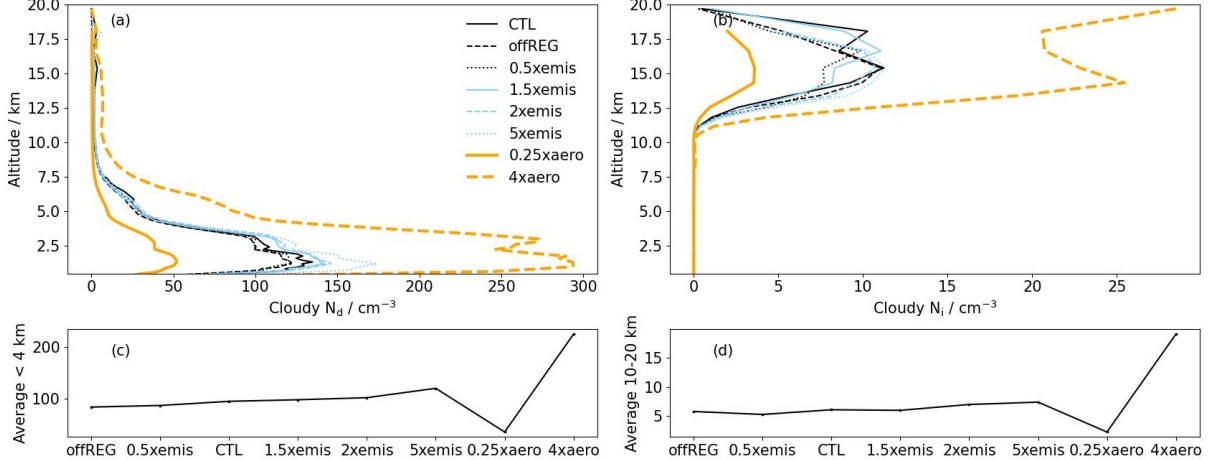

**Figure 7.** Profiles of (a) $N_d$ and (b) $N_i$, averaged over time and over the cloudy area of the high-sulfur region in the CTL (black solid), offREG (black dashed), 0.5×emis (black dotted), 1.5×emis (light blue solid), 2×emis (light blue dashed), 5×emis (light blue dotted), 0.25×aero (thick orange solid line) and 4×aero (thick orange dashed) simulations (upper panel). The lower panel is the $N_d$ (c) averaged over the lowest 4 km altitude of the profiles in the upper panel for the eight simulations, and $N_i$ (d) averaged between 12-20 km altitude of the profiles in the upper panel for the eight simulations. The results are from the 3-hourly instantaneous model output.

The in-cloud ice number concentration ($N_i$) is negligible from the surface to around 11 km in altitude, from which height it increases and peaks at around 15 km (11 cm$^{-3}$ in the CTL simulation). Changing the anthropogenic emissions in the regional

domain does not have a clear effect on $N_i$ between 12-20 km. Averaged over height between 12-20 km, $N_i$ in the simulations with the six emission scenarios have similar values (roughly 6 cm$^{-3}$) and the differences are negligible. However, in the 0.25×aero simulation, $N_i$ is reduced by a factor of 3 compared to the CTL simulation and in the 4×aero simulation it is increased by a factor of 3.

The profiles of in-cloud liquid and ice mass mixing ratios averaged over the high-sulfur regions exhibit several peaks at 3 km, 6

km and around 13 km in altitude for the eight simulations (Fig. 8). The cloud water is liquid phase below 4 km altitude, mixed phase between 4 and 10 km, and ice phase above 10 km altitude (Fig. A6). Cloud liquid water mass mixing ratio is similar among the eight simulations and it quickly increases with altitude from 1 km to 3 km reaching a maximum (0.46 g kg$^{-1}$), then decreases with height. Some clearer (but still not obviously systematic) differences among the eight simulations are shown for cloud ice mass mixing ratio which exists above about 5 km altitude, allowing the mixed-phase cloud to reach 0.6 g kg$^{-1}$ at

around 6 km and cloud ice mass to become 0.61 g kg$^{-1}$ at 14 km altitude. The results show that the variations of cloud ice with height are not affected by changes in anthropogenic emissions by factors of between 0 and 5 relative to the CTL simulation or when $N_d$ is significantly reduced or increased (0.25×aero and 4×aero).



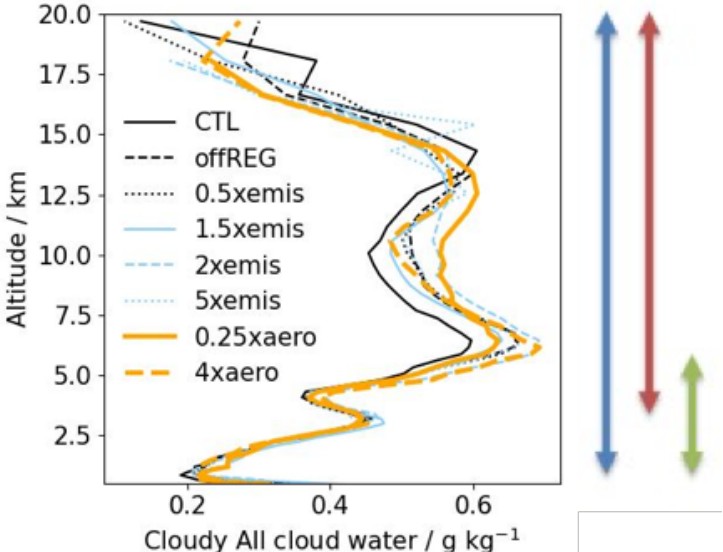

**Figure 8.** Profiles of total cloud water mass mixing ratio (cloud liquid, ice crystal, snow, and graupel), averaged over time and over the cloudy area of the high-sulfur region in the CTL (black solid), offREG (black dashed), 0.5×emis (black dotted), 1.5×emis (light blue solid), 2×emis (light blue dashed), 5×emis (light blue dotted), 0.25×aero (thick orange solid) and 4×aero (thick orange dashed) simulations. The results are from the 3-hourly instantaneous model output. The three arrows indicate the vertical extent of the cloud heights that we use to identify deep cloud (blue), shallow cloud at high altitude (red), and shallow clouds at low altitude (green) in Fig. 9.



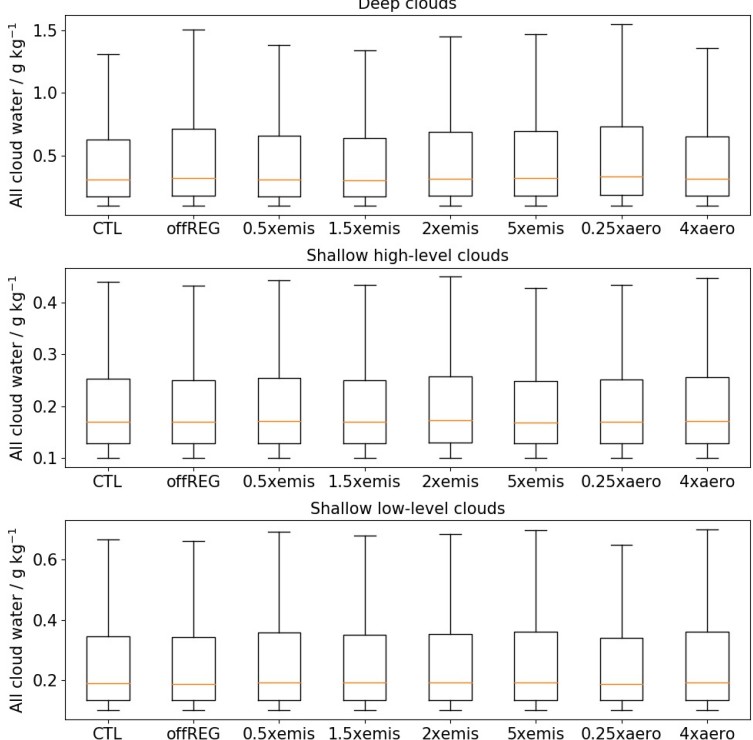

**Figure 9.** Box plots of total cloud water mass mixing ratios from all the 3-hourly instantaneous output in the cloudy area of the high-sulfur regions in the CTL, offREG, 0.5×emis, 1.5×emis, 2×emis, 5×emis, 0.25×aero and 4×aero simulations for deep clouds (cloud thickness greater than 3 km; top), shallow clouds at high altitude (cloud thickness smaller than 3 km and at above 4 km in altitude; middle), and shallow clouds at low altitude (cloud thickness smaller than 3 km and at below 5 km in altitude; bottom).

The distributions of total cloud liquid and ice mass mixing ratios are shown as box plots in Fig. 9 based on 3-hourly instantaneous output in the six emission scenarios, 0.25×aero and 4×aero simulations separated into deep clouds (thickness greater than 3 km), shallow clouds (thickness smaller than 3 km) situated below 5 km altitude, and shallow clouds situated above 4 km altitude. All three cloud categories have the same cloud water mass mixing ratio for the lower whiskers (minimum; 0.1 g kg$^{-1}$). For deep clouds, the lower whisker, lower quartile (the edge of the lower 25 %) and median values in the eight simulations generally differ by less than 0.03 g kg$^{-1}$, while the upper quartile values differ by at most 0.08 g kg$^{-1}$ and upper whisker values differ by 0.2 g kg$^{-1}$. The maxima for the upper quartile (the edge of the upper 75 %) and upper whisker (maximum) for deep clouds occur in the offREG and 0.25×aero simulations which have the least cloud droplets due to the least number of CCN. The water content in shallow clouds at low and high altitudes similarly shows no systematic dependence on aerosol concentrations. Overall the box plots show that the occurrence of 'extreme' values is random under varied anthropogenic emissions.





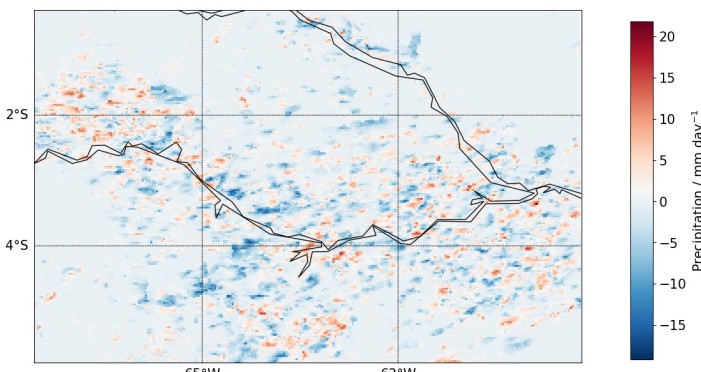

**Figure 10.** The map of the differences between CTL and 0.25×aero simulations for the time-mean surface rain rates in all the high-sulfur regions between 12 and 18 March 2014 with 3-hourly mean model output.

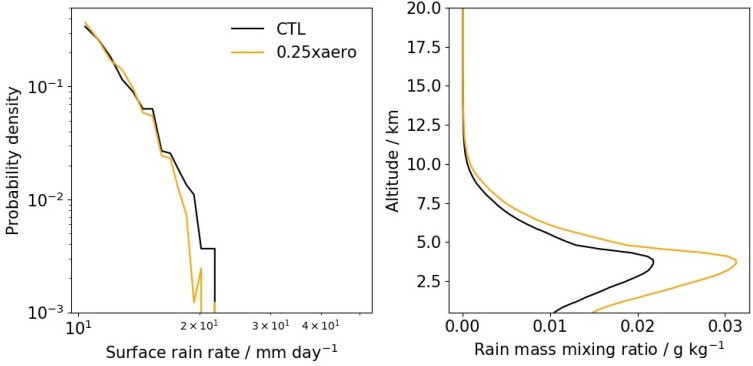

**Figure 11.** Histograms of surface rain rate (left) and profiles of rain mass mixing ratio (right) in high-sulfur regions in the CTL (black) and 0.25×aero (orange) simulations. The results are from the 3-hourly mean model output. The area under each line equals to 1.

A maps of differences of surface rain rate between the CTL and 0.25×aero simulations are shown in Figure 10 averaged over the
simulation period 12-18 March 2014. Because the high-sulfur regions evolve with time, the map shows all the locations where rain rate has ever occurred in the regions that meet the 'high-sulfur' threshold during the 7-day simulations. The perturbations to surface rain mostly occur close to the Amazon river where cities are located. Averaged over time, the surface rain in high-sulfur regions is increased by 0.16 mm day$^{-1}$ in the 0.25×aero simulation from the CTL simulation (4 % increase).

The distributions of surface rain rate in the CTL and 0.25×aero simulations are shown in Fig. 11. The histograms of surface
rain rate differ between CTL and 0.25×aero only for the upper end of the distribution above 16 mm day$^{-1}$. Similarly, the histograms of surface rain mass mixing ratios for all the eight simulations (Fig. A7) show that the changes are clear only for the maximum values (greater than 2 g kg$^{-1}$), while light rain is rarely affected. Although the differences in surface rain rate seem small between the CTL and 0.25×aero simulations, the profiles of rain mass mixing ratio in Fig. 11 and Fig. A6 show




that 0.25×aero exhibited at least twice as much as the change in other simulations vs the CTL simulation. The changes in rain
mass mixing ratio in the 0.25×aero from the CTL simulation are statistically significant (p value is 0.04). Therefore, rain is
only appreciably affected when the total aerosol number concentration is reduced significantly (0.25×aero).

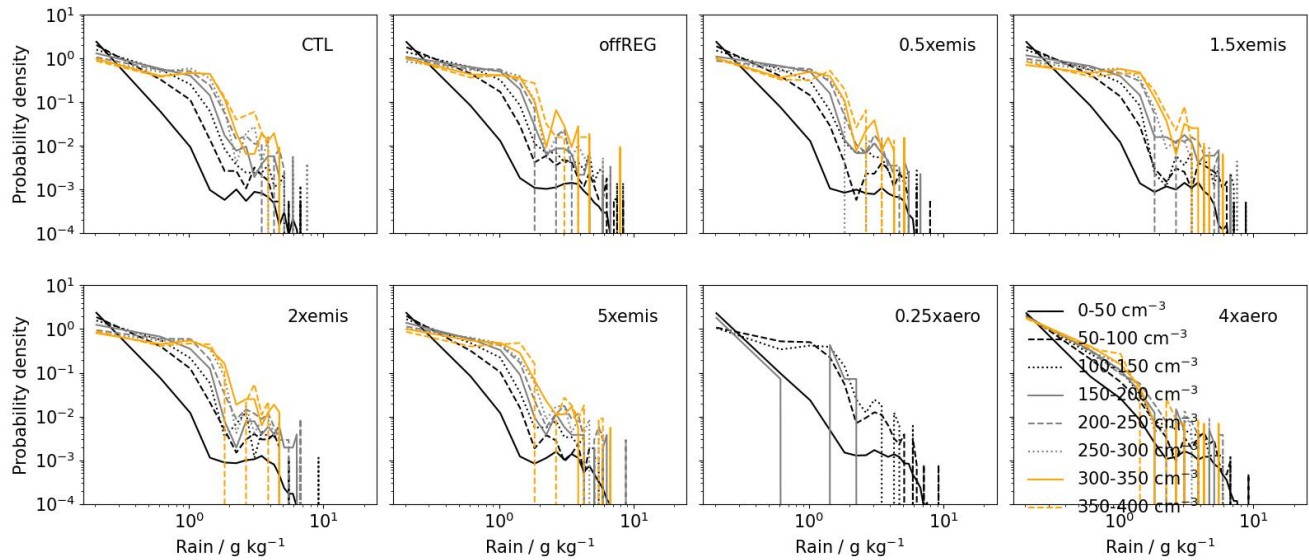

**Figure 12.** Histograms of surface rain mixing ratio in the high-sulfur regions in the CTL, offREG, 0.5×emis, 1.5×emis, 2×emis, 5×emis,
0.25×aero and 4×aero simulations. Rain results are separated into several column-mean cloud droplet number concentration bins for each
simulation (0-50 cm$^{-3}$ in black solid, 50-100 cm$^{-3}$ in black dashed, 100-150 cm$^{-3}$ in black dotted, 150-200 cm$^{-3}$ in grey solid, 200-250
cm$^{-3}$ in grey dashed, 250-300 cm$^{-3}$ in grey dotted, 300-350 cm$^{-3}$ in yellow solid, and 350-400 cm$^{-3}$ in yellow dashed). The results are
from the 3-hourly instantaneous model output. The area under each line equals to 1.

Surface rain mass mixing ratios in each simulation are decomposed into several column-mean cloud droplet number concentra-
tion bins in Fig. 12 to understand the relationship between cloud droplet concentrations and rain. The probability of a surface
rain mass mixing ratio smaller than 0.4 g kg$^{-1}$ decreases as cloud droplet concentration increases, while the probability of the
rain between 0.4 and 3 g kg$^{-1}$ tends to be larger as cloud droplet concentrations increase. It implies that with relatively light
to moderate rain ($< 0.4$ g kg$^{-1}$) a higher droplet number concentration suppresses rain, while a high cloud droplet number
concentration is necessary to generate or sustain a heavier rain (0.4-3 g kg$^{-1}$). Such effects are less significant in the 4×aero
simulation compared to the others. The probability of rain becomes similar in different cloud droplet number concentration
bins, i.e. rain is suppressed because of too many droplets formed from aerosols.

To improve the statistical significance of any changes, two 1-month CTL and offREG simulations were run from 11 March to
10 April 2014 (Fig. A8). The results are similar to the six 1-week simulations in that N$_{\text{d}}$, ice and liquid cloud mass mixing
ratio and rain mass mixing ratio are not significantly different between the CTL 1-month and offREG 1-month simulations.
For example, the differences in N$_{\text{d}}$ between the CTL 1-month and offREG 1-month simulations are 10 cm$^{-3}$ (10 % of the





CTL 1-month simulation) when averaged over time, height below 10 km altitude, and cloudy area of the high-sulfur regions.

The mean difference for $N_i$ above 10 km altitude is -0.2 cm$^{-3}$ (-31 %) and for the total cloud mass mixing ratio at all altitudes the difference is -0.03 g kg$^{-1}$ (-7.2 %). Rain mass mixing ratio differences are 0.002 g kg$^{-1}$ (16 %) below 10 km altitude in the high-sulfur regions. The histograms of surface rain mass mixing ratio in the polluted regions in the two simulations show that rain mass only differs from the other simulations when rain is greater than around 3 g kg$^{-1}$ (Fig. A7) with more high rain mixing ratios in the month long simulations. This is likely because the longer sampling time allows the occurrence of more

extreme rain rates.

## 4   Discussion and conclusions

We investigated the influences of anthropogenic emissions on aerosol particles, clouds and rain in central Amazonia using a regional model nested within a global atmosphere-only model, and we perturbed the regional domain with several high spatial resolution anthropogenic emission scenarios relative to a control simulation. The baseline simulation (CTL) compared well

with the observations in the areas where G-1 aircraft flew (mostly below 2 km) and the model captures the variability across the plume transects. Upper tropospheric nucleation, along with subsequent downward transport, has been shown to be important for determining low-level particle concentrations (Clarke et al., 1998, 1999a, b; Clarke and Kapustin, 2002; Pierce and Adams, 2007; Merikanto et al., 2009; Wang and Penner, 2009; Weigel et al., 2011; Wang et al., 2016; Williamson et al., 2019), and it is important for Amazonia during the dry season (Andreae et al., 2018). However, in our test simulations, switching

on upper tropospheric nucleation caused an overestimation of particle concentrations compared to the observations which are available mostly below 2 km altitude and occasionally at 2-6 km altitude. We therefore disabled nucleation above 1 km to achieve consistency between the model and observations in March 2014.

Switching off anthropogenic emissions in the regional domain (CTL to offREG simulation) caused reductions of aerosol number concentrations along the flight track by around -70 % to -90 % of the $N_{D>3nm}$, -50 % to -70 % for $N_{D>10nm}$, and reductions

of up to -20 % for $N_{D>100nm}$ particles along the flight tracts. The reductions in aerosol came from reductions in both primary and nucleated particles with the latter being the dominant source of the changes. Primary aerosol had a very small contribution to the smallest particles ($N_{D>3nm}$) and it contributed to around 10 % and less than 20 % of the $N_{D>10nm}$ and $N_{D>100nm}$ particles, respectively. Overall it contributed to around 0.5 % of the height-mean total particle concentrations below 4 km altitude in the polluted regions (Fig. 4). In this study, both the pure biogenic nucleation mechanism and the nucleation mechanism

that additionally uses $H_2SO_4$-Org to create new particles were used. The simulations showed that it is mainly the $H_2SO_4$-Org nucleation rate that responds to the changes in anthropogenic emissions ($SO_2$ that forms $H_2SO_4$) at lower altitudes after we prevented the upper tropospheric nucleation. The $H_2SO_4$-Org nucleation mechanism in our study was therefore the more important factor in controlling the particle concentration variations along the flight tracks, though the contribution would be smaller if upper tropospheric nucleation was included. The enhancement of particle number concentrations in Amazonia due

to anthropogenic emissions was also found in Shrivastava et al. (2019) and Zhao et al. (2021).





To quantify the effects of anthropogenic emissions on aerosol, cloud and rain, we focused on the regions that are strongly affected by anthropogenic emissions in the regional domain (termed high-sulfur regions) defined as an instantaneous column-integrated sulfur content (from $H_2SO_4$ and $SO_2$) below 2 km altitude that exceeds $6 \times 10^{-5}$ g m$^{-2}$ in the CTL simulation. We then compared the changes in aerosol, cloud and rain properties among the simulations in the high-sulfur regions. The

high-sulfur regions are dependent on the time and intensity of emissions as well as the wind fields.

For each unit increase in anthropogenic emissions in the regional domain (e.g. from CTL to 2×emis), the equivalent total aerosol number concentrations in the high-sulfur region increased by approximately 13 % averaged over time. The positive relationship between aerosol and anthropogenic emissions has also been found from some observational studies in which days of clean and polluted air in Amazonia were compared (Martin et al., 2016, 2017) and has also been found in the modelling

studies (Shrivastava et al., 2019; Zhao et al., 2021).

In the high-sulfur regions, we then analysed the relationship between particle concentration, nucleation rate, condensation sink and sulfuric acid with the column integrated sulfur content in the lowest 2 km altitude. Similar to the domain- and time-mean profiles, anthropogenic emissions enhance the $H_2SO_4$-Org nucleation rates and particle concentrations for all size ranges. However, the particle concentrations do not increase monotonically with the increasing column integrated sulfur content. The

nucleation mode aerosol, Aitken mode aerosol, $H_2SO_4$-Org nucleation rate and sulfuric acid concentration reach a plateau and subsequently have a reduction of around a factor of 4 at the sulfur content range between 4 and 5 g m$^{-2}$. This reduction is because of the increasing primary aerosol emission near the source of the pollution. The extra primary aerosols can act as a sink for sulfuric acid and subsequently suppress nucleation and accelerate coagulation, resulting in lower nucleation and Aitken mode aerosol concentrations. Zhao et al. (2021) also showed that nucleation was suppressed near the pollution source

in Manaus. The suppression on condensable gases, nucleation and tiny aerosols is most evident near the source of the pollution. The biogenic nucleation rate exhibits a slight decreasing trend with sulfur content and level of anthropogenic emissions due to the corresponding increase in condensation sink. Additionally, the lack of upper tropospheric nucleation prohibits the majority of biogenic nucleation, making it a less significant factor in influencing particle concentrations in Amazonia under this model setup (Merikanto et al., 2009; Kirkby et al., 2016; Gordon et al., 2016; Wang et al., 2023). However, it is important to take the

upper tropospheric biogenic nucleation into account in future studies to better understand the sources of aerosol particles in Amazonia.

We also investigated the influences of anthropogenic emissions on clouds. In the lowest 4 km altitude, the cloudy $N_d$ increased by 9 % for each unit increase of anthropogenic emissions. Higher anthropogenic emissions resulted in more cloud droplets because greater aerosol concentrations can produce more CCN which subsequently enhance cloud droplet concentrations (Zhao

et al., 2006; Kawamoto and Suzuki, 2012; Polonik et al., 2020; Pöhlker et al., 2021). Reducing the aerosol concentration caused a reduction of $N_d$ by a factor of 2 in the 0.25×aero simulation which reduced the aerosol concentration by a factor of 4 compared with the CTL simulation. Similarly, increasing the aerosol concentration by a factor of 4 results in more than doubled $N_d$. For ice particle number concentrations ($N_i$), we only found reductions in the 0.25×aero simulation and increase



in the 4×aero simulation which were caused by changes in $N_d$, and allowed different number droplets to freeze, while the rest of the simulations had similar $N_i$ because they had similar $N_d$. The relationship between cloud droplets and ice may have been weakened because heterogeneous ice nucleation was not interactively correlated with aerosol concentrations in the model. However, $N_d$ is likely to affect the number concentration of ice crystals through the homogeneous freezing of droplets. The correlation between $N_i$ and $N_d$ in the 0.25×aero and 4×aero simulations is consistent with previous studies that have shown that ice concentrations are affected by cloud droplet concentrations (Fan et al., 2013; Herbert et al., 2015; Grabowski and Morrison, 2020).

Our simulations explored how changes in aerosol affected cloud and rain water mass mixing ratios. The responses of total cloud water and rain mass mixing ratios were not statistically different among the various perturbation simulations. This absence of significant effects from aerosol may be explained by the multiple complex processes of aerosol-deep convection interactions that can buffer the effects of aerosol concentration perturbations. Connolly et al. (2013) stated that aerosols affected deep convective clouds in a non-linear way which caused complex changes of cloud and rain. Similar non-linear relationships have been addressed by Ekman et al. (2007) and van den Heever and Cotton (2007). The lack of substantial effects may also be associated with missing or over-simplified process representation in our cloud microphysics scheme, for example the lack of prognostic supersaturation (Fan et al., 2018). Additionally, Furtado and Field (2022) constructed an universal distribution for the rainfall fluctuations that is unlikely to be affected by aerosol perturbations when the rainfall is segregated by $N_d$. Their results imply that the effects of aerosol may influence the rainfall distribution but aerosols will not significantly change the distribution.

In contrast, reducing and increasing the concentrations of aerosol by a factor of 4 in the activation process (0.25×aero and 4×aero simulations) produced changes in $N_d$ and $N_i$ (relative to the baseline CTL simulation) that were at least a factor of 2 greater than in the other emission scenarios. The mean rain rate in 0.25×aero was then increased by around 4 % relative to the CTL simulation in the high-sulfur regions, but the histograms of rain rate did not show significant differences between the CTL and 0.25×aero simulations. The much greater response of rain in the 0.25×aero simulation implies that the perturbations to aerosol and $N_d$ were not large enough in the six anthropogenic emission scenarios to have triggered significant changes. However, the 4×aero simulation also shows insignificant changes in rain when there are many more background aerosols compared to the CTL simulation and the rain is suppressed for all the cloud droplet number concentration bins (Fig. 12). This pattern shows the non-linearity of rain as cloud droplet concentration increases: rain has already been suppressed as much as it can be at the aerosol concentrations of the offREG simulation, which may explain the lack of change in rain at higher aerosol concentrations.

Review studies (Rosenfeld et al., 2008; Tao et al., 2012; Fan et al., 2016) have highlighted the potentially complex relationships among aerosols, clouds, and precipitation, and similar messages have been conveyed by some modelling studies although the focus was not on the environment in Amazonia (Seifert et al., 2012; Fan et al., 2016; Alizadeh-Choobari, 2018; Barthlott et al., 2022; Furtado and Field, 2022). For example, Alizadeh-Choobari (2018) investigated mid-latitude cloud systems and pointed out that aerosols could cause a redistribution of rain and that the response of rain to aerosol loadings depended on rain



intensity. Barthlott et al. (2022) used the ICON model and found that the microphysical effects of higher CCN caused narrower cloud droplet distributions and reduced rain rates over Germany. However, using the COSMO weather forecast model, Seifert et al. (2012) found that aerosols had a negligible effect on surface precipitation over Germany. Evaporation was shown to be
enhanced with more aerosols because of the formation of more smaller sized cloud droplets which may subsequently release aerosols to the atmosphere (Leung et al., 2023). However, shallow cumulus clouds were found to be more sensitive to such enhanced evaporation than large congestus clouds because large congestus clouds are more likely to go through warm-phase invigoration rather than enhanced evaporation (Leung et al., 2023).

Overall, the relationships between anthropgoenic emissions, aerosols, clouds and rain are complex and no clear correlations
were found for changes in cloud liquid water, cloud ice water or rain to any changes in anthropogenic emissions. The insensitivity is potentially because the environment (even when switching off all anthropogenic emission) already has a lot of background aerosols even without anthropogenic emissions in the regional domain and the small perturbations of $N_d$ as a result of small changes in aerosols. However, we found distinct responses of clouds when the number of aerosols was directly reduced by a factor of 4 with a subsequent reduction in $N_d$ by a factor of 2. This indicates that local anthropogenic emissions
do not exert a strong control over CCN and cloud droplet concentrations within convective clouds over the scale of the regional domain, although it is possible that the anthropogenic emissions might have more impact further downwind since this would allow more time for growth of nucleated aerosols to CCN sizes (Wang et al., 2023).

The limitations of this study lie in the missing upper tropospheric NPF mechanism, simplified warm cloud microphysics, and missing aerosol interactive heterogeneous ice nucleation microphysics. We manually prevent NPF outside of the layer between
100 m and 1 km in altitude so that the regional model has a better representation of the observed aerosol particle concentrations. The compromise inhibits the descent of newly formed aerosols from the upper troposphere and their possible interactions with deep convection in the free and upper troposphere (Ekman et al., 2004; Yin et al., 2005; Fan et al., 2018). The simplified warm cloud microphysics and lack of aerosol-dependent heterogeneous ice nucleation might prevent liquid and ice water content changes in response to aerosol concentration changes. Thus, this study may not fully represent the response of cloud to
changes in aerosol. We recommend future studies to investigate how the background aerosol particles affect the aerosol-cloud interaction in this region by removing anthropogenic emissions globally. It is also recommended that future studies focus on the response of a single cloud to anthropogenic emissions in order to better understand the physical processes of the affected cloud, in a similar way to the study of Miltenberger et al. (2018b) which developed an ensemble to evaluate the response of cloud properties. Nevertheless, our study provides a more detailed understanding from a modeling perspective of the effects of
anthropogenic emissions and NPF to CCN and cloud droplet concentrations in the Amazonia wet season.

*Data availability.* The observations have been obtained from GoAmazon2014/5 campaign (Martin et al., 2016, 2017). Raw model data are available through JASMIN service (http://www.jasmin.ac.uk/, last access: 9 January 2025). We have uploaded a subset of simulation data that were used to produce the figures to Zenodo (doi:10.5281/zenodo.7213371).



# Appendix A

**A1**

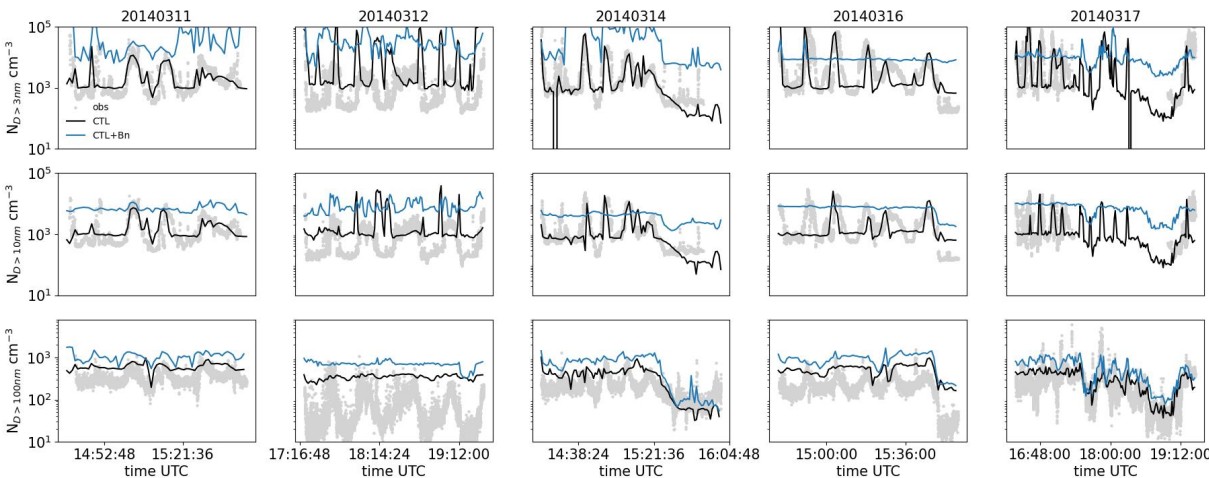

**Figure A1.** Time series of observed (grey dots) and modelled (CTL and CTL+Bn; solid lines) particles number concentrations with diameters greater than 3 nm (upper row), 10 nm (middle row) and 100 nm (lower row) on 11, 12, 14, 16 and 17 March 2014. The observations were measured onboard the G-1 aircraft during GoAmazon2014/5 campaign and model data are interpolated according to the G-1 flight tracks. Black solid lines are for the CTL simulation, and blue lines are for the CTL+Bn simulation where binary nucleation ($H_2SO_4$-$H_2O$) is switched on above 100 m altitude.

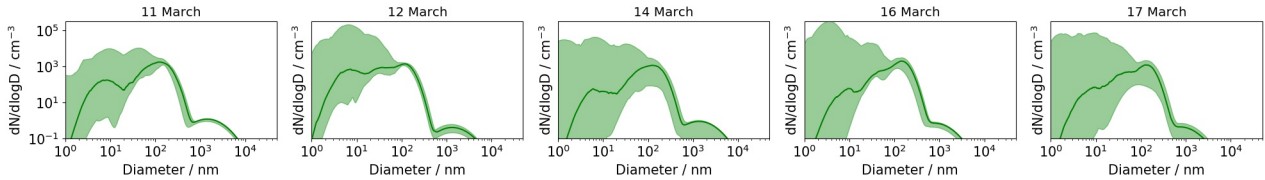

**Figure A2.** The medians of particles size distributions on 11, 12, 14, 16 and 17 March 2014 in the CTL simulation. The model data have been interpolated according to the time, coordinates and altitudes of the G-1 flight tracks. Shades represent the 97.5 % and 2.5 % of the distributions at all the interpolated time.



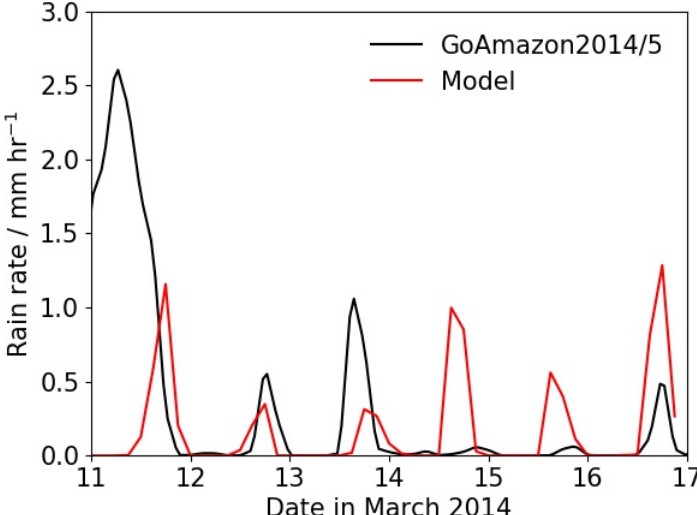

**Figure A3.** Precipitation rate observed by S-band Amazon Protection National System radar at 3.2°S, 60.6°W during GoAmazon2014/5 from 11 to 17 March 2014 (black) and precipitation rate from the model in CTL simulation (red) averaged over the radar domain (approximately 2° by 2°W domain that centres at 3.2°S, 60.6°W).





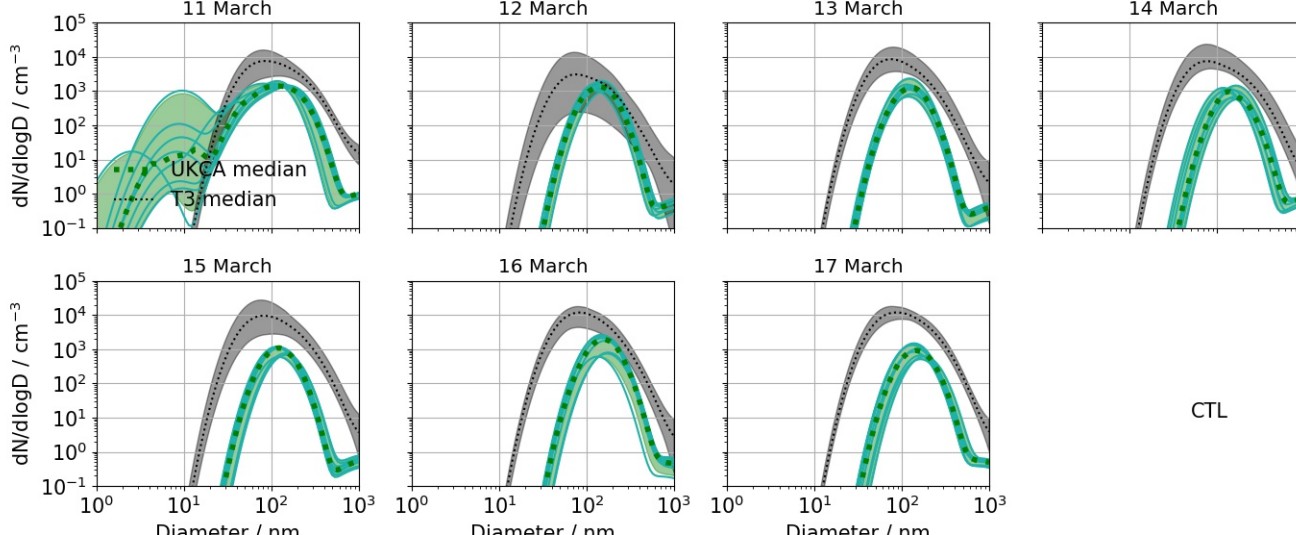

**Figure A4.** Aerosol size distributions from 11 to 17 March in the CTL simulation (light green lines and green dotted lines) and measured at T3 research tower (black dotted lines; 3.2°S, 60.6°W) for the aerosols with diameters between 55 nm and 1000 nm. The light green lines indicate individual 3-hourly instantaneous model output for each day and the green dotted lines are the medians of the instantaneous result for each size bin. The observations have time resolution of 10 seconds and the black dotted lines are the medians of the observations for each size bin. The shaded grey and green area are the 97.5 % and 2.5 % percentiles for the observations and the CTL simulation.



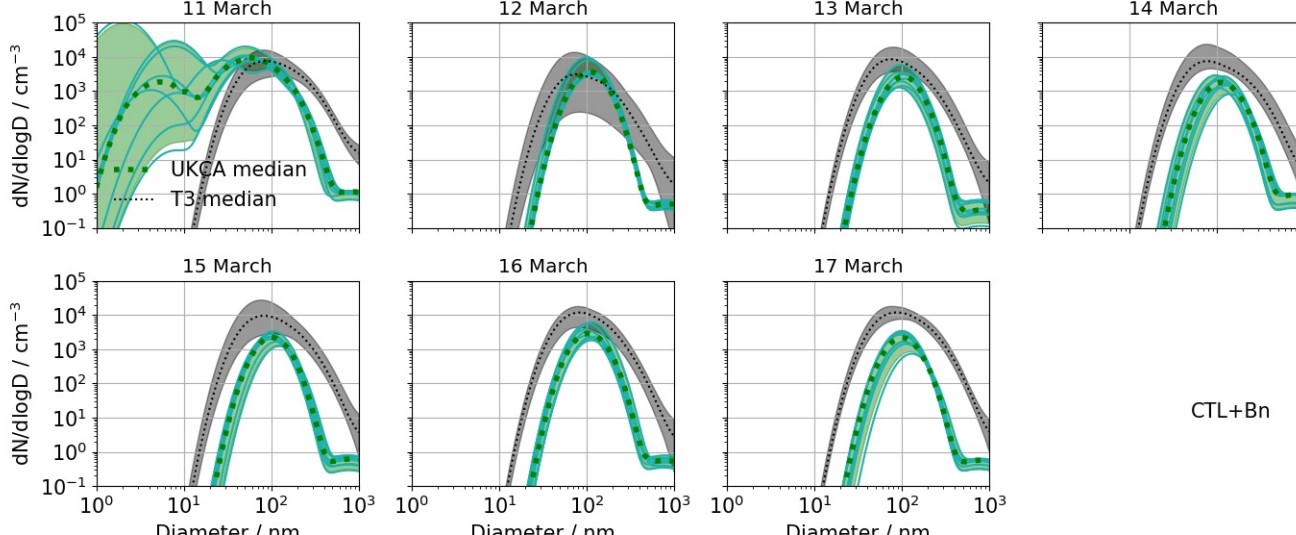

**Figure A5.** Aerosol size distributions from 11 to 17 March in the CTL+Bn simulation (light green lines and green dotted lines) and measured at T3 research tower (black dotted lines; 3.2°S, 60.6°W) for the aerosols with diameters between 55 nm and 1000 nm. The light green lines indicate individual 3-hourly instantaneous model output for each day and the green dotted lines are the medians of the instantaneous result for each size bin. The observations have time resolution of 10 seconds and the black dotted lines are the medians of the observations for each size bin. The shaded grey and green area are the 97.5 % and 2.5 % percentiles for the observations and the CTL+Bn simulation.





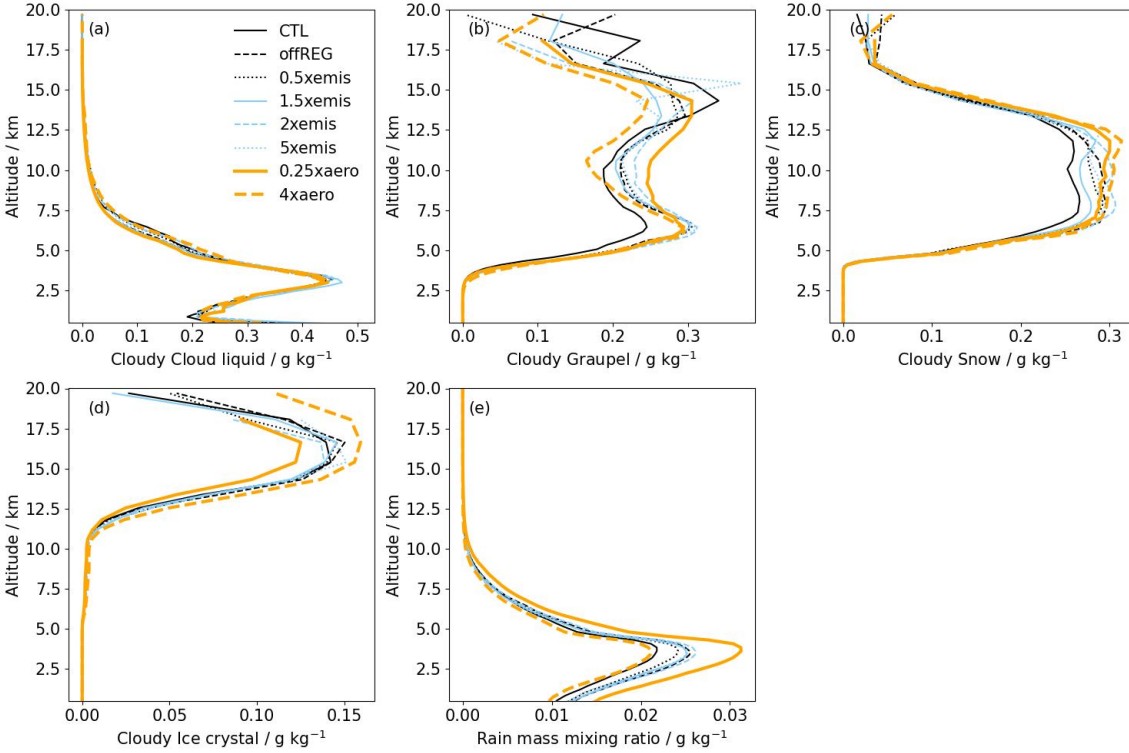

**Figure A6.** Profiles of (a) cloud liquid, (b) graupel, (c) snow and (d) ice crystal mass mixing ratio, averaged over time and over the cloudy area of the high-sulfur region in the CTL (black solid), offREG (black dashed), 0.5×emis (black dotted), 1.5×emis (light blue solid), 2×emis (light blue dashed), 5×emis (light blue dotted), 0.25×aero (thick orange solid) and 4×aero (thick orange dashed) simulations. Profiles of (e) rain mass mixing ratio are averaged over time and the area of the high-sulfur region for the eight simulations.



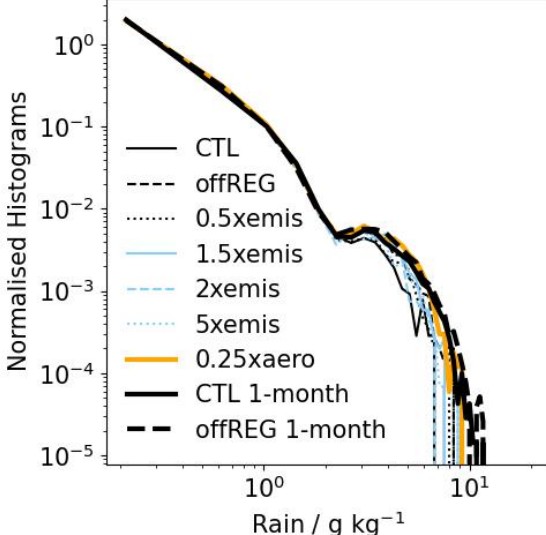

**Figure A7.** The histograms of surface rain mass mixing ratios in high-sulfur regions in the CTL (black solid), offREG (black dashed), 0.5×emis (black dotted), 1.5×emis (light blue solid), 2×emis (light blue dashed), 5×emis (light blue dotted), 0.25×aero (thick orange solid) and 4×aero (thick orange dashed) simulations. The figure also includes the CTL (thick black solid) and offREG (thick black dashed) simulations that have been run for 1 month. The area under each line equals to 1.



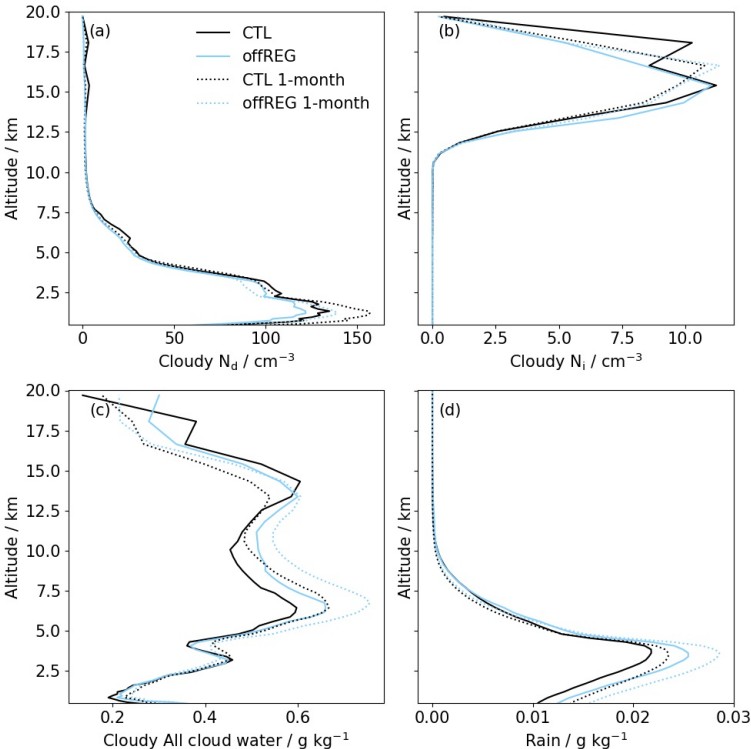

**Figure A8.** Profiles of (a) $N_d$, (b) $N_i$ and (c) total cloud water mass mixing ratio, averaged over time and over the cloudy area of the high-sulfur region in the CTL (solid black) and offREG (solid light blue) simulations that are run for 7 days (solid), and two extra simulations that are run for 1 month, CTL 1-month (dotted black), and offREG 1-month (dotted light blue). Profiles of rain mass mixing ratio (d) are averaged over time and the area of the high-sulfur region for the four simulations.

*Author contributions.* XW, KSC, DPG and HG designed and led this research. The regional configuration of UM-UKCA was provided by HG and DPG. HG provided the codes for the inorganic-organic nucleation mechanism and helped produce the model-observation comparison (Fig. 2). XW ran the model simulations, analysed the model results, and wrote the paper with insights, comments, and edits from KSC, HG and DPG.

*Competing interests.* One of the authors is a member of the editorial board of Atmospheric Chemistry and Physics.

*Acknowledgements.* The research was funded by Marie Skłodowska-Curie and was with the CLOUD-MOTION project (grant 764991). KC acknowledges support from the Natural Environment Research Council (NERC), the Aerosol-Cloud Uncertainty REduction project (A-CURE) under the grant number NE/P013406/1. HG acknowledges support from the US Department of Energy's Atmospheric System Re-



search Program under grant number DE-SC0022227. We acknowledge the UK Met Office for providing the support for UM-UKCA-CASIM and Monsoon Superco(o)mputing Node to run the simulations and the JASMIN team with whose platform we processed our modelled data. We also acknowledge the Atmospheric Radiation Measurement (ARM) database for providing the observations under the GoAmazon2014/5 campaign and thank Jennifer Comstock for advising on the observational data. We thank Ananth Ranjithkumar for helping identify the oxidant fields for the nucleation mechanism of the coupled chemistry.




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
