# Peer review of "Weak influence of anthropogenic emissions on aerosol, cloud and rain in the wet season of the Amazon rainforest"

_EGUsphere, 2025_

## Referee Comment (RC2)

**Review for ACP manuscript EGUSPHERE-2025-132.**

The influence of Amazonian anthropogenic emissions on new particle formation, aerosol, cloud, and surface rain. From Xuemei Wang1,2, Ken S. Carslaw1, Daniel P. Grosvenor1,3, and Hamish Gordon2

This manuscript, EGUSPHERE-2025-132, addresses crucial processes that connect particle formation in the remote tropical atmosphere, particle growth, and CCN concentrations in Amazonia. This manuscript investigates the impact of anthropogenic emissions from Manaus, Brazil, on new particle formation (NPF), aerosol concentrations, cloud properties, and rainfall in the Amazon region. The authors used the HadGEM3 climate model as a nested regional climate model to simulate various emission scenarios and analyze the effects of these emissions. The use of a high-resolution nested regional domain (3 km resolution) in the HadGEM3 model enables a highly precise representation of local atmospheric processes and their interactions with anthropogenic emissions. The study highlights the complex interactions between anthropogenic emissions, aerosol formation, and cloud processes in this relatively pristine environment, but very significantly, regionally and globally.

In terms of methodology, the use of a high-resolution regional model nested within a global model allows for detailed simulations of aerosol and cloud processes. The study design, with its various emission scenarios, is effective in isolating the impact of anthropogenic emissions.  The inclusion of observational data from the GoAmazon2014/5 campaign provides a strong basis for model validation. Comparing model results with aircraft measurements and radar data enhances the credibility of the findings. The paper is generally well-structured, with a clear introduction, detailed methods section, well-organized results, and a comprehensive discussion. The figures and tables are effective in presenting the data.

The authors acknowledge several limitations in their model, including the exclusion of upper tropospheric NPF and simplifications in cloud microphysics. These limitations may affect the accuracy of the simulated aerosol-cloud interactions. However, it is well recognized that NPF in high altitudes is a very recent finding. In all climate and regional models, cloud microphysics must be simplified to a great extent.

Some of the emission scenarios, particularly those involving very strong reductions in aerosol number concentrations, are considered unrealistic. While these scenarios help to explore the system's sensitivity, their relevance to real-world conditions is limited. The complex nature of aerosol-cloud-precipitation interactions leads to some difficulties in interpreting the model results. The authors acknowledge the nonlinearities and buffering effects in the system, which can make it challenging to draw definitive conclusions.

Future work should focus on incorporating more comprehensive representations of cloud microphysics and aerosol-cloud interactions. Including upper tropospheric NPF would provide a more complete picture of aerosol formation and transport. **:** The study could benefit from a more in-depth analysis of the mechanisms driving the observed relationships between anthropogenic emissions, aerosols, and cloud properties. For example, a more detailed examination of the role of specific chemical species and microphysical processes would be valuable.

The finding that even drastic reductions in aerosol concentrations lead to only a 4% increase in rainfall raises questions about the model's sensitivity. This outcome may suggest limitations in the model's ability to capture the complexities of cloud and precipitation processes in a non-linear convective environment. I think a more comprehensive discussion on this aspect could be good for the manuscript.

**Overall Recommendation:**

This is a well-executed and valuable study that contributes to our understanding of the complex interactions between anthropogenic emissions and the Amazonian "natural" atmosphere. The authors use appropriate methods and present their data clearly. While the model has some limitations and challenges in interpreting the results, the study offers valuable insights into the complex interactions between anthropogenic emissions, aerosols, and cloud processes in the Amazon. I believe that the most valuable finding in this study is that the Amazonian climate-aerosol-CCN system is **highly resilient to change** (see the first specific comment). I recommend that the title could include the high resilience of the system.

**I recommend that ACP accept the manuscript for publication, after responding to the specific questions listed below.**

**Some specific comments.**

**Abstract:** It focuses on how resilient the Amazonian system is: "The 7-day simulations show that, in the areas that are affected by anthropogenic emissions, when aerosol and precursor gas emissions are doubled 10 from the baseline emission inventories, aerosol number concentrations increase by 13 %.". "We also found that doubling the anthropogenic emission can increase the cloud droplet number concentrations (Nd) by 9 %". "Even extreme reductions in aerosol number concentrations by a factor of 4,

which is an unrealistic condition, cause only 4 % increase in rain over the domain." This could be somewhat included in the article's title.

**Introduction line 20:** Are you sure that for this statement, you need 14 references at the same point? Maybe choosing one or two as the best reference would be better. This is also true for lines 34, 52, and others. Far too many citations that are not actually relevant to the manuscript.

Section 2.2 - Line 150: "CMIP6 emission inventories provide CH4, and monoterpene, isoprene and natural SO2 are from CMIP5 inventories." I think we need more details on the isoprene emission inventory since it plays a critical role in natural aerosol production in Amazonia.

Section 2.2 - The manuscript does not mention the natural primary biogenic aerosol particles. This is a crucial component of the Amazonian background aerosol, and it must be considered. How was it treated in the manuscript?

Section 2.2 – Anthropogenic SO2 is obtained from Edgar. What about the sulfate precursors that could come from DMS and sulfur compounds from flooded areas?

Section 2.3 – line 180-185: Monoterpenes are the main VOCs for boreal forests, but NOT for Amazonia. Isoprene chemistry is the primary particle precursor in Amazonia, as reported by numerous studies spanning the lower troposphere to the high troposphere, as shown in the Curtius et al. paper from 2024.

Section 2.3 - What about NPF driven by isoprene-NOx system? The GoAmazon papers show that the increase in ozone is driven by NOx emissions, which have significant consequences for aerosol production.

Results line 255: It is not true that the model represents N > 100 well, as can be seen on 20140312. These are logarithmic plots, so there are differences by a factor of 10. In the legend, make clear that the plots are on a logarithmic scale.

Section 4 discussions and conclusions - Line 570 – Your strong statement "Thus, this study may not fully represent the response of cloud to changes in aerosol". Needs better qualification, as this was one of the study's primary objectives. On line 84, you explicitly mention the objective: "(2) What are the mechanisms that drive changes in aerosol and cloud properties?".

---

## Author Response (AR1)

**Reply to the comments for egusphere-2025-132**

We thank the reviewers for their valuable comments and suggestions which have helped improve the quality of the manuscript. The reviewer's comments are shown below in black, and our responses are provided in blue. New sentences and paragraphs have been added to the manuscript, highlighted in green italics. Line numbers refer to those in the revised manuscript. In addition to the changes made in response to the reviewers' comments, we revised the entire manuscript to improve the overall flow. Detailed changes can be found in the attached diff file for all the tracked changes.

We also fixed some formatting bugs related to DOIs with double 'https://doi.org/' in the References section.

**Reply to the first referee**

This manuscript, EGUSPHERE-2025-132, addresses crucial processes that connect particle formation in the remote tropical atmosphere, particle growth, and CCN concentrations in Amazonia. This manuscript investigates the impact of anthropogenic emissions from Manaus, Brazil, on new particle formation (NPF), aerosol concentrations, cloud properties, and rainfall in the Amazon region. The authors used the HadGEM3 climate model as a nested regional climate model to simulate various emission scenarios and analyze the effects of these emissions. The use of a high-resolution nested regional domain (3 km resolution) in the HadGEM3 model enables a highly precise representation of local atmospheric processes and their interactions with anthropogenic emissions. The study highlights the complex interactions between anthropogenic emissions, aerosol formation, and cloud processes in this relatively pristine environment, but very significantly, regionally and globally.

In terms of methodology, the use of a high-resolution regional model nested within a global model allows for detailed simulations of aerosol and cloud processes. The study design, with its various emission scenarios, is effective in isolating the impact of anthropogenic emissions. The inclusion of observational data from the GoAmazon2014/5 campaign provides a strong basis for model validation. Comparing model results with aircraft measurements and radar data enhances the credibility of the findings. The paper is generally well-structured, with a clear introduction, detailed methods section, well-organized results, and a comprehensive discussion. The figures and tables are effective in presenting the data.

The authors acknowledge several limitations in their model, including the exclusion of upper tropospheric NPF and simplifications in cloud microphysics. These limitations may affect the accuracy of the simulated aerosol-cloud interactions. However, it is well recognized that NPF in high altitudes is a very recent finding. In all climate and regional models, cloud microphysics must be simplified to a great extent.

Some of the emission scenarios, particularly those involving very strong reductions in aerosol number concentrations, are considered unrealistic. While these scenarios help to explore the system's sensitivity, their relevance to real-world conditions is limited.

The complex nature of aerosol-cloud-precipitation interactions leads to some difficulties in interpreting the model results. The authors acknowledge the non-linearities and buffering effects in the system, which can make it challenging to draw definitive conclusions.

Future work should focus on incorporating more comprehensive representations of cloud microphysics and aerosol-cloud interactions. Including upper tropospheric NPF would provide a more complete picture of aerosol formation and transport. : The study could benefit from a more in-depth analysis of the mechanisms driving the observed relationships between anthropogenic emissions, aerosols, and cloud properties. For example, a more detailed examination of the role of specific chemical species and microphysical processes would be valuable.

The finding that even drastic reductions in aerosol concentrations lead to only a 4% increase in rainfall raises questions about the model's sensitivity. This outcome may suggest limitations in the model's ability to capture the complexities of cloud and precipitation processes in a non-linear convective environment. I think a more comprehensive discussion on this aspect could be good for the manuscript.

Overall Recommendation:

This is a well-executed and valuable study that contributes to our understanding of the complex interactions between anthropogenic emissions and the Amazonian "natural" atmosphere. The authors use appropriate methods and present their data clearly. While the model has some limitations and challenges in interpreting the results, the study offers valuable insights into the complex interactions between anthropogenic emissions, aerosols, and cloud processes in the Amazon. I believe that the most valuable finding in this study is that the Amazonian climate-aerosol-CCN system is highly resilient to change (see the first specific comment). I recommend that the title could include the high resilience of the system.

I recommend that ACP accept the manuscript for publication after responding to the specific questions listed below.

We thank the reviewer for providing detailed explanations and suggestions. We have addressed the model's simplifications and their potential impact on the representation of the aerosol-cloud-precipitation interactions in the Discussion and Conclusion section (see line 646)

 "The study could benefit from a more in-depth analysis of the mechanisms driving the observed relationships between anthropogenic emissions, aerosols, and cloud properties. For example, a more detailed examination of the role of specific chemical species and microphysical processes would be valuable."
Regarding the mechanisms behind the relationships (quote paragraph above), we now state that our study provides insights into the response of aerosols, cloud properties and precipitations to changes in anthropogenic emissions in a small region. However, it is limited by a few simplified or missing processes in the model. Adding in-depth

investigation of specific microphysical processes would help better understand the mechanism. For example, when we scaled the number of aerosols in 0.25xaero and 4xaero simulations, the resulting changes in rain formation were small. It implies that cloud and rain water are buffered by factors such as background meteorology. We also analysed the relationship between gas-phase sulfur content, which is directly related to particle formation, with several variables in high-sulfur regions shown in Fig. 5 and 6. Such analysis is helpful in understanding the environment, but due to the limitations of the current model setup (e.g. resolution, bulk rather than finely resolved sectional aerosol and cloud microphysics), some more in-depth analysis cannot be performed. Other variables that we could look at, for example liquid or ice water path are closely associated to cloud water content which has shown to not be sensitive to perturbations in anthropogenic emissions. Chemical species, except for those containing sulfur, are not directly related to particle formation and will probably not have a big influence on CCN concentrations. We do not expect them to add more values to the study and thus are not included. Therefore, we added to the recommendation that future studies may consider using a high-resolution cloud-resolving model or a large-eddy simulation.

The following sentence has been added to line 646 "*The study provides insights into the response of aerosols, cloud properties, and precipitation to changes in anthropogenic emissions in a small region, but it is limited by some simplified or not included processes in the model.*" The recommendation has been added to line 650 "*It is also recommended that future studies focus on the response of a single cloud to anthropogenic emissions using a higher resolution (e.g. large-eddy simulation) in order to better understand the physical processes of the affected cloud*" and line 653 "*Additionally, having a thorough investigations of the influences of cloud microphysical processes (e.g. ice formation, autoversion and accretion) on cloud and rain properties will improve our understanding of the complex environment.*"

For the comment: "The finding that even drastic reductions in aerosol concentrations lead to only a 4% increase in rainfall raises questions about the model's sensitivity. This outcome may suggest limitations in the model's ability to capture the complexities of cloud and precipitation processes in a non-linear convective environment. I think a more comprehensive discussion on this aspect could be good for the manuscript." Fig. 10 in the manuscript shows that precipitation increases and decreases in a patchy way and it results in an overall increase of 0.16 mm/day from CTL to 0.25xaero. Paca et al. (2020) showed trend of precipitation for over 37 years and the trends are more organized than our Fig. 10 but still has great spatial variability (with a mean of 2.8 mm/year, a maximum of 45.1 mm/year and minimum of -37.9 mm/year). It suggests that changes in precipitation are likely to occur in relatively small scales which contributes to the 'patchy' pattern, and such pattern is more significant in our 7-day runs.

The relatively small changes in precipitation despite large reductions in aerosol number concentrations may also reflect limitations in the model's ability to capture the complex and non-linear convective processes. For example, 3 km resolution is a factor to limit the performance of the model. 3 km resolution will not fully resolve all convection in the domain. Another example is that cold rain process will likely be limited in the current

model setup. Ice formation depends on temperature and is moderated by cloud droplet number concentrations. We added the following to the discussion and conclusion section at line 588: "*The 3 km resolution does not resolve all convection in the model and the transport of heat and moisture may be limited at smaller scales. The current temperature-dependent ice formation scheme, which is not aware of aerosol particles, will limit the model's ability to simulate cold rain*". We also added to recommendations that future studies increase the horizontal resolution to investigate the influence of anthropogenic emissions on cloud and rain properties (see line 650).

Some specific comments.

Abstract: It focuses on how resilient the Amazonian system is: "The 7-day simulations show that, in the areas that are affected by anthropogenic emissions, when aerosol and precursor gas emissions are doubled 10 from the baseline emission inventories, aerosol number concentrations increase by 13 %.". "We also found that doubling the anthropogenic emission can increase the cloud droplet number concentrations (Nd) by 9 %". "Even extreme reductions in aerosol number concentrations by a factor of 4, which is an unrealistic condition, cause only 4 % increase in rain over the domain." This could be somewhat included in the article's title.

We have revised the title "*Weak influence of anthropogenic emissions on aerosol, cloud and rain in the Amazon rainforest*".

Introduction line 20: Are you sure that you need 14 references at the same point for this statement? Maybe choosing one or two as the best reference would be better. This is also true for lines 34, 52, and others. There are far too many citations that are not relevant to the manuscript.

We reduced the number of citations and removed some citations that are not relevant to this study for lines 23 and 50.

There was one citation at line 34 in the first submitted manuscript, but we went through all the text and reduced the number of citations at lines 27, 58, 515, and 565.

Section 2.2 - Line 150: "CMIP6 emission inventories provide CH4, and monoterpene, isoprene and natural SO2 are from CMIP5 inventories." We need more details on the isoprene emission inventory since it plays a critical role in natural aerosol production in Amazonia.

In our simulations, monoterpenes and isoprene are obtained from the monthly mean emission inventories generated by the JULES model (Pacifico et al., 2012). While isoprene is important in NPF in the Amazon, our model configuration does not simulate HOM formation from isoprene with NOx, which is closely related to isoprene-driven NPF. We then do not incorporate isoprene in the NPF process in our study. However, our assumption for monoterpenes (instead of isoprene) to be the main NPF precursor gas is expected to be within the uncertainty. We have addressed this limitation in in the manuscript. Please also see explanations for the comment 'Section 2.3 – line 180-185' in page 6 in this document.

We added the following sentences to explain isoprene emission and the reason why isoprene is not incorporated in the NPF process. We have also addressed this limitation in the discussion.

Line 152: "*Monoterpenes and isoprene are emitted by vegetation and have been obtained from monthly mean emission inventories generated by the JULES model (Pacifico et al., 2012)*".

Line 166: "*Recent work has suggested that isoprene is an important BVOC involved in NPF (Kuhn et al., 2010, Bardakov et al., 2024, and Curtius et al., 2024, Shen et al.. 2024). However, isoprene is not used in the NPF process in this work because HOM formation from isoprene with NOx is not available in our model configuration, and this fairly new NPF mechanism has not been parameterised or tested in global models (Curtius et al., 2024, Shen et al.. 2024).*"

Section 2.2 - The manuscript does not mention the natural primary biogenic aerosol particles. This is a crucial component of the Amazonian background aerosol, and it must be considered. How was it treated in the manuscript?

There are natural marine organic aerosols and sea salt that can be advected into the Amazon rainforest. We added an additional paragraph to describe these aerosols at line 162 "*Our model also includes natural primary aerosol (sea salt and primary marine organic aerosol). The parameterisation of sea salt aerosols follows Gong et al. (2003), while primary marine organic aerosol emissions are based on Gantt et al. (2012). Dust emission is parameterised based on Marticorena et al., (1995)*".

Aerosols from biomass burning and biofuel combustion are treated as anthropogenic aerosols. Descriptions of these aerosols are provided in section 2.3.

Other aerosols such as pollen, bacteria and spores are not included in the model. Similar to Zhao et al (2022 GRL), we are not aware of reliable emissions estimates for primary biological particles. Also, Heald and Spracklen (2009) suggests that they may be a major contributor to aerosol mass concentrations, and later research suggests they are important INP, but they are likely a relatively small contributor to CCN, our focus here, due to their generally lower number concentrations Pöschl et al. (2010). The natural variabilities of these aerosol particles may influence the representation of the observed $N_{d>100nm}$. We have added the limitations later in the discussion section that missing some natural primary aerosols may have contributed to the discrepancies in simulating $N_{d>100nm}$ at line 636.

Section 2.2 – Anthropogenic SO2 is obtained from Edgar. What about the sulfate precursors from DMS and sulfur compounds from flooded areas?

DMS and sulfate compounds from flooded areas are not included in this model. Sulfate from DMS in the flooded regions has the potential to influence secondary aerosol formation but the past studies have not shown whether it can significantly affect CCN. We acknowledge this limitation and have added the following text in the discussion and conclusion section at line 636: "*Additionally, including extra primary aerosols (such as pollen, bacteria and spores), as well as sulfate compounds in the flooded areas that may lead to secondary sulfate production, could improve the representation of $N_{D>100nm}$ in the Amazon. However, these primary biological aerosol particles may have limited*

*impact on cloud droplet number concentration due to their low concentrations in this region and their role in activating aerosols to form cloud droplets (Heald and Spracklen., 2009, Pöschl et al., 2010). Andreae et al. (1990) has shown that although the forest has a large sulfur gas emission, the concentrations of aerosol are more mainly associated with marine and anthropogenic sulfate aerosol. Other relevant studies have not quantified the contribution of sulfur from floodplain to secondary aerosol formation (Brinkmann and Santos, 1974, Andreae and Andreae, 1988, Jardine et al. 2015)."*

Section 2.3 – line 180-185: Monoterpenes are the main VOCs for boreal forests, but NOT for Amazonia. Isoprene chemistry is the primary particle precursor in Amazonia, as reported by numerous studies spanning the lower troposphere to the high troposphere, as shown in the Curtius et al. paper from 2024.

We agree that monoterpene is not the main VOC contributing to NPF in the Amazon as shown in Curtius et al. (2024), Shen et al. (2024) etc. Due to the limitations of the chemical mechanism that does not incorporate HOM formation from isoprene with NOx, and the fact that this NPF parameterization has not been implemented or tested in global models, isoprene driven NPF could not be represented and thus isoprene is not considered in NPF in our study. We then assume that monoterpene is the primary BVOC that is responsible for NPF in our simulations. In this and our previous study (Wang et al. 2023), we investigated the effect of different monoterpene oxidation rates on aerosol population in the Amazon. The results have been validated using aircraft measurements (particle concentrations) and we believe that using monoterpene instead of incorporating isoprene will be within the uncertainty of what we assume for monoterpenes. We have revised the manuscript as follows.

The relevant sentence at line 165 has been moved to section 2.3 and has been revised:
*"Monoterpenes are a class of BVOC (Biogenic Volatile Organic Compound) consisting of several compounds, but they are emitted and treated as one tracer in the UKCA model. We assume it to be the main BVOC for biogenic nucleation. Recent work has suggested that isoprene is an important BVOC involved in NPF (Kuhn et al., 2010, Bardakov et al., 2024, and Curtius et al., 2024, Shen et al.. 2024). However, isoprene is not used in the NPF process in this work because HOM formation from isoprene with NOx is not available in our model configuration, and this fairly new NPF mechanism has not been parameterised or tested in global models (Curtius et al., 2024, Shen et al.. 2024). Nevertheless, not incorporating isoprene-NOx is expected to be within the uncertainty of our assumption for monoterpenes."*.

Section 2.3 - What about NPF driven by isoprene-NOx system? The GoAmazon papers show that the increase in ozone is driven by NOx emissions, which have significant consequences for aerosol production.

NPF driven by isoprene-NOx is not included in our model configuration which does not represent HOM formation from isoprene with NOx. We stated this limitation in the discussion and conclusion section. We added the following text to discussion. The studies by Kuhn et al. (2010), Bardakov et al. (2024), Curtius et al. (2024), and Shen et al. (2024) have emphasized the potential importance of isoprene in particle formation in the tropical forests. This model configuration does not incorporate such processes which may contribute to uncertainties (but not much) in particle concentrations greater than 100 nm diameter. We have added the following text to the manuscript at line 509:

*"However, the model sometimes does not reproduce the magnitude and temporal variability for particles greater than 100 nm. Possible reasons are listed below:*

- *(1) It may be related to the absence of some primary sources such as natural pollen, or additional anthropogenic emissions from the Manaus region.*
- *(2) Upper tropospheric nucleation (including NPF from the organic compounds formed by isoprene with NOx (Kuhn et al., 2010, Bardakov et al., 2024, Curtius et al., 2024, and Shen et al., 2024)), along with subsequent downward transport, has been shown to be important for determining low-level particle concentrations (Clarke et al., 1998, Clarke et al., 1999, Clarke et al., 2002, Merikanto et tal., 2009, Wang et al., 2016, Williamson et al., 2019, Curtius et al., 2024), and it is important for Amazonia during the dry season (Andreae et al., 2018), but we expect the absence of isoprene-NOx mechanism to be within the uncertainty of current NPF mechanism"*.

And at line 635: *"Not including isoprene and nitrates in NPF may introduce some inconsistencies between the simulations and the real world."*

Although the absence of some processes may limit the model's performance, we believe that it is not expected to change our final conclusion that states the resilience of cloud and rain water to change in anthropogenic emissions. We also added to recommendation that future studies are encouraged to incorporate isoprene driven NPF to represent the aerosol-cloud-precipitation interactions in the Amazon according to the most recent studies. A new sentence has been added to line 654 *"Parameterising isoprene nitrates nucleation based on the most recent results is recommended for future studies in this region."*

Results line 255: It is not true that the model represents N > 100 well, as can be seen on 20140312. These are logarithmic plots, so there are differences by a factor of 10. In the legend, make clear that the plots are on a logarithmic scale.

We agree that the model does not represent $N_{D>100nm}$ well, especially on 12 March. To address this point, we revised the sentences to more accurately refelct the model as follows at line 293 *"The CTL simulation reproduces most of the observed in-plume number concentrations for $N_{D>3nm}$ and $N_{D>10nm}$, and the general trend for $N_{D>100nm}$, except for 12 March. However, the magnitude of concentrations and the temporal variability is not well captured in the first 3 days (11, 12 and 14 March 2014) for $N_{D>100nm}$."*

We also clarified it in the figure caption that the concentrations in Fig. 2 are presented in a logarithmic scale.

Section 4 discussions and conclusions - Line 570 – Your strong statement "Thus, this study may not fully represent the response of cloud to changes in aerosol". Better qualifications are needed, as this was one of the study's primary objectives. On line 84, you explicitly mention the objective: "(2) What are the mechanisms that drive changes in aerosol and cloud properties?".

Thank you for pointing it out! We wanted to state that due to the limitations we have mentioned in our study (e.g. model resolution, simplified processes in chemistry and cloud microphysics), there is still room for improvements. We agree that the original sentence was too strong and have revised it.

The results show that anthropogenic emissions influence aerosol particles and subsequently cloud droplet concentrations. However, even when cloud droplet concentrations are greatly changed, changes in cloud water content are still not significant. Precipitation is also rather resilient to changes in aerosol and cloud droplet number concentrations. Although there are uncertainties in the chemistry scheme, we do not expect the main result to be significantly different. We have revised the sentence at line 646 to "*The study provides insights into the response of aerosols, cloud properties, and precipitation to changes in anthropogenic emissions in a small region, but it is limited by some simplified or not included processes in the model. Nevertheless, we do not expect these limitations to significantly affect our conclusions*".

**Reply to the second referee**

This manuscript presents a modelling investigation into the role of anthropogenic pollution from Manaus, Brazil, on CCN, clouds and precipitation downwind in Amazonia. They do this based on 5 days of simulations compared to measurement data collected from aircraft during the GoAmazon campaign in March 2014/15 (wet season). The setup includes a regional model embedded in a global model with horizontal resolution of 3 km, making it suitable for explicit representation of deep convection. They present sensitivity studies and their implications for particle number concentration, droplet and ice crystal concentrations and rainfall. Overall, they conclude that the impact of regional emissions from Manaus is moderate in terms of both aerosol number and cloud droplet number concentration and low for cloud water mass and rain. The paper is mostly well written, interesting and it is well within the scope of ACP.

I have two more general comments and then detailed comments below:

- Overall, the analysis in the paper focuses on high sulfur regions. I would expect NPF to have a larger impact in the regions where the plume is diluted somewhat into the cleaner surroundings and I am wondering why the research question is limited to the in-plume conditions? I am not suggesting that this needs to be changed, but rather that this could be discussed.

  We agree that pollution can have a large impact on a clean region. When pollution is slowly transported and diluted into the downwind areas, it can influence particle concentrations in the surrounding clean air. We analysed the ratios of Aitken mode to insoluble Aitken mode aerosol number concentrations as a function of distance from Manaus (the pollution source) at around 550 m altitude for the 6 experiments with scaled emission loadings (see Fig. 1 in this document). We assume the Aitken mode aerosol as an indicator of NPF and particle growth, and insoluble Aitken mode aerosol as representative of primary emissions. The figure shows that NPF and growth are suppressed within ~20 km downwind of Manaus, while the ratio increases significantly around at distances of 100-200 km from Manaus in all the 6 simulations. It suggests a stronger influence of anthropogenic emissions on particle formation and growth in

regions farther downwind before the pollution is completely diluted or scavenged. This figure has been included in the manuscript as Fig. A11.

[Figure]

Fig. 1 The ratio of Aitken to insoluble Aitken mode aerosol number concentration against the distance from Manaus in CTL, offREG, 0.5xemis, 1.5xemis, 2xemis and 5xemis simulations.

However, in our study, the high-sulfur regions are not fixed temporally or spatially, but evolve with the plume. As a result, the low-sulfur regions are usually not significantly influenced by anthropogenic emissions and always have low pollution levels. We have also checked the results in low-sulfur regions, and found that the responses of particles and clouds to changes in anthropogenic emissions are negligible. We analysed the relationship between aerosol concentrations with gas-phase sulfur content (Fig. 5) which shows the influence of gas-phase sulfur as it gradually increases. It shows similar results (to Fig. 1 in this document) that nucleation rates are enhanced in moderate sulfur while very high sulfur does not enhance nucleation.

To clarify this point, we have added a new paragraph to the discussion section at line 605 "*Analyses in regions with the column integrated gas-phase sulfur content lower than $6x10^{-5}$ g m$^{-2}$ are not included in this study. These regions generally exhibit minimal sensitivity to the perturbations in anthropogenic emissions (see Fig. A10) because all regions that are potentially affected by anthropogenic emissions are already included in the high-sulfur regions. As the high-sulfur regions vary with time, the remaining areas with gas-phase sulfur content lower than the threshold are usually not affected by the anthropogenic plume and therefore are not the focus of the analysis*".

- This model description section (2.2.) doesn't read very well in my opinion. There's quite a bit of repetition and the information is not well structured. For the aerosol description specifically, I would suggest having a separate paragraph on BVOC oxidation products and yields, then introduce the different nucleation rates (is it correct that there is no sulfuric acid-water nucleation rate btw?), then

say something about the overestimation in the models and where you turn off the nucleation rate.

Thank you for the helpful advice. We have revised section 2.2 to improve the content. The repeated sentences have been removed and this section has been organised as follows:

- o 2.2 Global and regional model configuration
- o 2.3  Aerosol and chemistry
- o 2.4 New particle formation
- o 2.5 Coupling between aerosol and cloud microphysics (moved from section 2.2).

The descriptions of monoterpene, HOM1 and HOM2 have been moved to the end of Section 2.3 as a separate paragraph to clarify the oxidation pathways.

For nucleation, sulfuric acid-water nucleation ($H_2SO_4$-$H_2O$), is not included in the main analysis. We tested it separately by switching on the $H_2SO_4$-$H_2O$ nucleation in the CTL simulation and found that it overestimated the particle concentrations in the free troposphere (Fig. A1 and A5). Although new particle formation in the upper troposphere has been found important in Amazonian dry season (Andreae et al., 2018), we could only compare GoAmazon2014/5 (wet season) which focused on particles in the lower atmosphere. Based on the CTL+Bn simulation, we also manually decreased the binary nucleation rate by a factor of 100 for an additional test which still overestimated the particle concentrations. These results have been added to Fig. A1.

We added new text to explain the overestimations of particle concentrations at line 190 "*In our test simulations with $H_2SO_4$-Org and pure biogenic nucleation mechanisms, the total particle number concentrations in the free troposphere were overestimated by more than a factor of 10 if we allow NPF to occur at all altitudes. The overestimation was even stronger with binary nucleation ($H_2SO_4$-$H_2O$) and the $H_2SO_4$-Org nucleation schemes*". We also added a new sentence at line 198 to clarify that we limit NPF above 1 km and below 100 m for both $H_2SO_4$-Org nucleation and pure biogenic nucleation mechanisms.

The cloud microphysics scheme also deserves a bit more detail. How is rain initiated for example? Ice formation is mentioned in the discussion/results, but not in the description.

We have included additional information to describe the rain and ice formation in the CASIM model.

- Rain formation: Autoconversion and accretion follows the  Khairoutdinov and Kogan (2000) and the self-collection of rain droplets (with rain droplets ) and cloud droplets (with cloud droplets) is based on Beheng et al. (1994). We have added the following text to the manuscript at line 241 "*Rain formation (autoconversion and accretion) from cloud droplets follows Khairoutdinov and Kogan (2000). The self-collection of rain droplets (with rain droplets) and cloud droplets (with cloud droplets) is based on Beheng et al. (1994)*".

- Ice formation: Ice formation follows Cooper et al. (1986) and we added the following sentence to the microphysics section at line 239 "*We use a temperature dependent ice nucleation scheme, which is not sensitive to aerosol, to form ice in the CASIM model (Cooper, 1986)*".

L16: I think it would help if you specify what is meant by non-linear here.
We have clarified that non-linear describes that the changes in aerosol number concentrations do not lead to subsequent changes in cloud or rain properties. It suggests that other processes (e.g. meteorology) have greater impact on the cloud and rain and the convection itself is resilient to the changes we have made.

We revised the last two sentences in the abstract at line 16 as "*If we assume our simulation has fine enough grid resolution and an accurate representation of the relevant atmospheric processes, the simulated weak response of cloud and rain implies that their properties have a non-linear response to the changes in anthropogenic emissions and aerosol concentrations in the Amazonian convective environment. It also implies that the convective environment is resilient to the changes in $N_d$ that occur in response to localised anthropogenic aerosol perturbations*".

L58-59: The meaning is somewhat unclear to me here. Consider rewording.
The sentence has been rephrased and the new sentence at line 55 is now revised to "*The impact of aerosols on deep convective systems is overshadowed by strong large-scale meteorological forcing and dynamical feedbacks that appear to diminish aerosol-induced perturbations*".

L105-107: The sentence does not flow well; the order of information seems slightly off. I suggest splitting it in two with the first containing information about the height and the second about the number concentrations.
The sentence has been rephrased and split into 2. It is now changed into (line 102) "*During the 5 days, most of the measurements were made below 2 km altitude, with a small fraction collected between 2-6 km altitude. Below 2 km, the concentrations of particles with diameters greater than 3 nm is around 18000 $cm^{-3}$, while between 2 – 6 km, the concentration is significantly smaller (100 – 200 $cm^{-3}$) compared to below 2 km.*"

L127: Is 3-km resolution enough to resolve all types of convection? What about shallow convection?
3-km resolution can only partially resolve some convection events and fully resolving convection, especially shallow convection, would require a higher resolution (a few hundred meters). However, it is sufficient to represent the key features of deep convective systems, resolve heat transfer, and transport tracers, while allowing us to keep a relatively large domain. To clarify this point, the sentence has been revised to (line 125): "*The regional model uses explicit convection which allows heat transfer and tracer transport to be resolved on the model grid, though smaller scale convection (e.g. shallow convection) is not resolved at 3-km resolution.*"

L136-137: This is a bit unclear: "i.e. collision and coalescence of aerosol-containing cloud droplets with some of the aerosol assumed to be deposited to the surface."

We revised the sentence to more clearly express the information. It is as follows (line 134): "*Aerosol particles are scavenged by two processes: impaction scavenging due to precipitation (washout) below clouds and scavenging during rain formation (rainout). Rainout refers to the collision and coalescence of cloud droplets which contain aerosols. When these rain droplets are formed and fall to the surface, the aerosols inside are assumed to be deposited*."

L154: What does "that include vegetation" mean? Also, why are these not interactive? What about diurnal variability?
We are thankful for the comment. We revised the sentence for clarity. Line 152: "*Monoterpenes and isoprene are emitted by vegetation and have been obtained from monthly mean emission inventories generated by the JULES model*".

For not using interactive vegetation, our model is an atmosphere-only configuration of the HadGEM3 climate model with a nested regional domain. Coupling the model with an interactive land-surface model with interactive vegetation cover and BVOC emissions would be helpful, but its influences are likely limited under the scope of our study which focuses on the influenes of anthropogenic emissions. Additionally, the vegetation cover is unlikely to change within a short period. In our model setup, the monoterpene and isoprene fields have been generated by an interactive land-surface model JULES which can alternatively provide sufficiently good monoterpene and isoprene fields without a more interactive land-surface model.

For diurnal variability: In the model, isoprene has a diurnal variation that scales the emissions hourly. There is no diurnal variability of monoterpene in this model setup. The emission of monoterpene is updated every 5 days.

We added the following sentences to clarify the 2 points at line 154: "*We use offline isoprene and monoterpene emissions because our study mainly focuses on the influence of anthropogenic emissiosns and the vegetation cover is unlikely to change significantly within a short time period. The benefits of using a land-surface model with interactive vegetation cover and BVOC emissions would be helpful, but the benefits would be limited under the context of our study. Diurnal variability has been applied to isoprene emissions by scaling them hourly. We do not apply a diurnal cycle to monoterpenes fields*".

L155: Are all primary aerosols emitted at the same size?
Thanks for pointing it out. We have corrected the the emitted diameters as follows. Line 159: "*Primary biofuel aerosol, biomass burning aerosol, and anthropogenic sulfate aerosol are emitted in the UKCA model as lognormal modes with a fixed geometric mean diameter of 150 nm, while primary aerosol particles from fossil fuel are emitted at 60 nm*."

L171-172: "The ability of biogenic vapour to nucleate depends on vapour volatility." I feel like this should be followed by a statement about how the model treats the formation of ELVOCs/HOMs.

Thank you for your comment. In the early stage of this study and an earlier study (Wang et al., 2023), we tested the sensitivity of several oxidaton rates for monoterpenes (oxidation by OH and O3) to investigate their influence on the formation of new particles via high-oxidised molecules (HOMs). By changing the oxidation rates of monoterpenes which provides a hint for vapour volatility (e.g. for isoprene), we explored how particle population were affected. However, the analysis in this study does not include the work related to oxidation rates, we have decided to remove the sentence to avoid confusion.

L173: "NPF in the UKCA model produces aerosol particles up to 3 nm in diameter." Unclear what the implication is. Also, below you only mention parameterizations giving 1.7 nm particles. Do you mean that you parameterize growth up to 3 nm in diameter? Which parameterization do you use for this?

Thank you for pointing out the ambiguity. The NPF process in UKCA model includes the initial formation of critical cluster at 1.7 nm and the subsequent growth by condensation to 3 nm in diameter. We have clarified the sentence in the manuscript to better describe the process. We also added a new sentence to describe the method for particle growth from 1.7 mn to 3 nm. Line 186 is now changed to "*The NPF process in the UKCA model includes the initial formation of a cluster at a diameter of 1.7 nm and the subsequent growth to 3 nm by condensation (Kerminen and Kulmala, 2002). We apply the Kerminen and Kulmala (2002) method to simulate particle growth from 1.7 nm to 3 nm via the condensation of $H_2SO_4$-$H_2O$ and HOM1 (or HOM2). The whole NPF process produces aerosol particles up to 3 nm in diameter*".

L175: Why below 100 m altitude? It's only mentioned earlier that the model overestimates NPF in the free and upper troposphere?

We suppressed NPF below 100 m altitude because NPF was rarely observed at this altitude in the Amazon rainforest and the model produced too many particles from NPF at low altitudes, possibly because of limitations of the model's mixing scheme close to heterogeneous forest. It has been revised to (line 193): "*We also found significantly overestimated particle concentrations from NPF at low altitudes in our test simulations, possibly due to the limitations of the model's mixing scheme close to heterogeneous forest. Additionally, NPF was rarely observed in the Amazonian boundary layer in previous studies (Krejci et al., 2003; Rizzo et al., 2010; Andreae et al., 2018; Wimmer et al., 2018; Varanda Rizzo et al., 2018).*"

L178: I don't quite follow here. You find that even when you turn it off above 1 km and below 100 km, NPF is still the dominant source?

The sentence could be improved to better describe the information. We meant to describe that NPF is the main source of particles with diameters smaller than 100 nm (not for all sizes) in this regional domain. As this sentence was not clear, we have removed it to avoid confusion.

L181: This sentence seems a bit out of place. Move to earlier?

We agree that the sentence repeats the text at the beginning of this section and better placed earlier. It is now revised and we moved it to line 202. It is now as follows: "*The formation and the subsequent growth of new particles uses highly-oxygenated*

*molecules (HOM1) (Ehn et al., 2014; Kirkby et al., 2016; Tröstl et al., 2016; Stolzenburg et al., 2018; Bianchi et al., 2019) and H$_2$SO$_4$.".*

L184: The yields for HOM1 should be mentioned here?
We added the yield of HOM1 (100%). The unrealistically high yield is because the nucleation rate and yield could not be separately constrained in the chamber experiments and so the yield has been subsumed into the nucleation rate. The sentence is now updated as (line 174) "*HOM1 is an oxidation product of monoterpenes, oxidised by OH with a yield of 100% (Riccobono et al., 2014). The unrealistically high yield is because the nucleation rate and yield could not be separately constrained in the chamber experiments and so the yield has been subsumed into the nucleation rate*" It has been moved to the previous section 2.3.

L184: "Here, monoterpene is a type…" Everywhere, not just here :) monoterpene is a class of BVOCs, and I guess it is treated as one tracer by the model?
We agree that it was misleading. We have removed the word 'here' and stated that monoterpenes are a class of BVOCs but are treated as a tracer in the model. The sentence has been changed to (line 165) "*Monoterpenes are a class of BVOC (Biogenic Volatile Organic Compound) consisting of several compounds, but they are emitted and treated as one tracer in the UKCA model*". It has been moved to section 2.3.

L195: "HOM2 is oxidized […]" maybe instead "HOM2 is the oxidation product of monoterpene by OH and O3."
We have revised the sentence accordingly and it has been moved to section 2.3 at line 175. The sentence is now as follows: "*HOM2 is the oxidation product of monoterpenes by OH and O$_3$, and HOM2 concentrations are obtained by a steady-state approximation (Franchin et al., 2015; Gordon et al. 2016).*"

L195: "a steady-state approximation". Please explain.
We have now added an explanation is now added in the following sentence to section 2.3 at line 176 "*A steady state assumes that ion concentrations remain constant over time, given a fixed recombination coefficient, first-order loss term, and coagulation sink (Franchin et al., 2015). This approximation is based on the CLOUD chamber experiments*".

L197: Repetition.
Thank you for pointing it out. We have removed the repeated part "for particles at 1.7 nm in diameter following Gordon et al. (2016)" to reduce redundancy.

L204-208: Do you mean that you include this in your survival probability calculation to the smallest nucleation mode? Do you use Kerminen and Kulmala (2002, https://linkinghub.elsevier.com/retrieve/pii/S002185020100194X) here? Maybe state explicitly.
The condensable gases and newly formed particles are typically affected by condensation sink from background particles, but clouds are usually not considered in this process. However, due to the convective environment in Amazonia, the influence of clouds should be not be neglected. Therefore, we included a condensation sink term

derived from cloud droplets and ice crystals based on Wang et al. (2023). The sum of condensation sink from background particles and cloud particles is used to suppress NPF. We have revised the manuscript as follows:

- Line 219: "*The calculation of cloud condensation sink follows the study of Wang et al. (2023).*"
- We also added an extra sentence by the end of the paragraph at line 228. "*The cloud condensation sink is added to the condensation sink derived from background particles. The total condensation sink will more realistically influence the concentration of condensable gases and newly formed particles in this convective environment.*"

Kerminen and Kulmala (2002) is not used to derive cloud condensation sink. This method is applied only when particles are growing from 1.7 nm to 3 nm and we now add new sentences to describe the incorporation of Kerminen and Kulmala (2002) method at line 187.

L255-256: I would say maybe that for N100 the model mostly misses the rapid variability though? I.e. it only captures the large drops towards e.g. the end of the days some days.
Thanks for the comment. We have revised the sentences to better reflect it. Line 293: "*The CTL simulation reproduces most of the observed in-plume number concentrations for $N_{D>3nm}$ and $N_{D>10nm}$, and the general trend for $N_{D>100nm}$ except for 12 March. However, the magnitude of concentrations and the temporal variability is not well captured in the first 3 days (11, 12 and 14 March 2014) for $N_{D>100nm}$.*"

L270: Same as above.
Thanks for pointing it out. We have removed this sentence because it is a repetition of the previous one.

L272: I don't think you describe the CTL-Bn simulations and H2SO4-H20 nucleation above in the description. It's a bit unclear to me as well if H2SO4-H20 nucleation is only turned on in this simulation and if you have the same restrictions in the vertical (i.e. turned off above 1 km).
Thanks for pointing it out. We added the following text to clarify the setup of CTL+Bn simulation. Line 271: "*We ran an extra simulation (CTL+Bn) to examine the effect of binary nucleation ($H_2SO_4$-$H_2O$) following Vehkamaki et al. (2002). In this simulation, binary nucleation is switched on in addition to the processes used in the CTL simulation. As binary nucleation is most effective in the upper troposphere, it is permitted at all altitude above 100 m, and due to its strong temperature dependence it would be negligible below 100 m if it were permitted there.*"

L285: I assume this is gas phase only? In the sentence before you write that these regions are defined "define according to the total sulfur species" which could be thought to be gas plus particle phase. I suggest clarifying this.
Yes, it refers to the gas phase only. We added "gas-phase" to specify that the calculation of gas-phase sulfur content is based on gas-phase species only.

L336-339: I have a hard time following this. I suggest including an equation to make it clear.

We added an equation to explain how to derive it. Line 380 "*The calculation is as follows (Conc$_{5xemis}$-Conc$_{CTL}$)/(5-1) if taking 5xemis simulation as an example*". This is an example for the simulations 5xemis and CTL.

L381-382: In addition to H2SO4 and nucleation rate going down, condensation sink goes down as well. I would have thought that the H2SO4 and nucleation rate went down due to high condensation sink, but this cannot be the explanation here?

Yes, we agree that the results were not expected. We investigated a few things that may explain such results, and they are listed as follows.

- Model grids with the largest gas-phase sulfur content are likely where NPF is not permitted in the lowest 100 m:
  - The model grids that fall into the largest gas-phase sulfur content bin are likely very close to the emission source which are mostly below 100 m altitude. However, NPF at these altitudes are not permitted because observations have not found NPF close to the surface. Therefore, it results in smaller nucleation rates than model grids with slightly lower sulfur content.

- Sulfur content represents only the gas-phase $SO_2$ and $H_2SO_4$ and $SO_2$ dominates:
  - The sulfur content accounts for the gas-phase $SO_2$ and $H_2SO_4$ in the lowest 2 km altitude and does not include the particle-phase sulfur. Therefore, model grids with the highest $SO_2$ do not completely coincide with those with the highest aerosol concentration. It is possible that we may miss some of the anthropogenic emission influenced regions that have high primary particle emission. However, as our dynamic high-sulfur regions cover relatively large areas and because wet season usually has low biomass burning events, we expect this influence to be small.
  - We also analysed the correlation of $SO_2$ with gas-phase sulfur content, and it shows that the gas-phase $SO_2$ dominates the gas-phase sulfur content (compared to $H_2SO_4$). The figure showing the relationship between $SO_2$ and the gas-phase sulfur content is included in appendix (Fig. A6 in the revised version).
  - The oxidation of $SO_2$ to $H_2SO_4$ and the subsequent formation of particles will take at least a few hours. It partly explains the low condensation sink and $H_2SO_4$ concentration because of reduced or prohibited nucleation.

- Limited number of data in the highest gas-phase bin:
  - We examined the number of gas-phase sulfur content data points in the lowest 2 km altitude for each bin and found that the largest bin ($\geqslant 0.01$) accounts for only 0.00091% of the total number (68448000) of gas-phase sulfur content datapoints. The total number is calculated as follows 68448000=200*310*23*48, where 200 and 310 are the number of model grids in horizontal direction excluding the upwind of Manaus region, 23 is the number of vertical levels in the lowest 2 km altitude, and 48 is the number of 3-hourly output. The number of grids with gas-phase sulfur content greater than $1\times10^{-2}$ g m$^{-2}$ is very rare, making it fairly difficult to represent.

We edited this paragraph as follows (line 422) "*In the largest gas-phase sulfur content bin, the aerosol concentration of all sizes, nucleation rate, condensation sink, and sulfuric acid have significant reductions. It is related to several factors. Firstly, these data are collected very close to the pollution sources which are usually below 100 m where nucleation is not permitted, resulting in a reduction in nucleation rates. Secondly, the sulfur content is derived from the gas-phase $SO_2$ and $H_2SO_4$ with $SO_2$ being the dominant contributor (see Fig. A6), therefore the model grids with the highest $SO_2$ do not always coincide with those with the highest aerosol concentration. Also, as oxidation takes at least a few hours, $H_2SO_4$ or particles cannot be formed quickly very close to the source, partly resulting in low particle concentrations, $H_2SO_4$ and condensation sink. Thirdly, the largest bin contains less than 0.001 % of all data points in the lowest 2 km altitude across all time steps, which may not accurately represent the concentrations, nucleation rates and condensation sink. As a result, the model grids that fall into the largest gas-phase sulfur content bin ($\geq$ 1x10$^{-2}$ g m$^{-2}$) has the highest gas-phase $SO_2$ while the other 7 variables/tracers (particle concentrations, nucleation rates, condensation sink and $H_2SO_4$) have very low values.*"

We also revised the second sentence in the previous paragraph to clarify that the increases in primary aerosol concentration does not account for the significant reduction found in the largest gas-phase sulfur content bin (line 414) "*The reductions in this range (4x10$^{-3}$ - 1x10$^{-2}$ g m$^{-2}$) are partly due to the rapidly increasing primary aerosol emissions in this gas-phase sulfur content range, but primary aerosol does not explain the significant reduction in the largest gas-phase sulfur content bin ($\geq$ 1x10$^{-2}$ g m$^{-2}$; Fig. 5.b and c; yellow)*". The relevant text in the abstract has also been updated accordingly.

L402-403: Why are the responses to the 0.25xaero and the 4xaero so symmetrical? Would you not expect some leveling off of the impact at high aerosol concentrations due to entering the updraft limited regime?
The aerosol number concentrations in the two simulations (0.25xaero and 4xaero) are scaled based on the CTL simulation. We modified the variable "$N_i$" in Equation 13 of Abdul-Razaak and Ghan (2000). In the real world, the relationship between aerosol and cloud droplet number concentration should not be linear and the two simulations should not look symmetrical. In this way, the variable is changed after the maximum supersaturation is determined and as a result, the simulations do not reflect the updraft-limited conditions and exhibit a nearly symmetrical response of $N_d$ profiles in 0.25xaero and 4xaero simulations. Although this direct modification is not realistic, it allows us to quickly investigate the response of the cloud properties to a significant change in aerosol particle concentrations.

We added the following text to the end of the method section to clarify the changes. Line 261: "*Two additional simulations were performed in which the aerosol concentrations passed from UKCA to the CASIM aerosol activation process were scaled down by a factor 4 (simulation 0.25xaero) and up by a factor of 4 (simulation 4xaero) relative to the CTL simulation. The variable we scaled is the "$N_i$" in Equation 13 in the study of Abdul-Razaak and Ghan (2000), where "i" represents an index over the aerosol modes. The scaling is applied after aerosol concentrations are passed into the CASIM*

*model from the UKCA model, thus influencing the cloud activation process without changing the aerosol number concentrations generated by the UKCA model*".

Fig. 8: The resolution of the arrows is quite bad.
We have produced the figure again with higher resolution.

L417-419: I suggest checking if these differences are significant and if not potentially removing this discussion.
We calculated the P-value using a T test by comparing CTL simulation against each of the other simulations. The P-values are very small (<0.00001) for deep convective clouds although it may be related to large dataset (more than 800000 data points). We revised these two lines as follows (line 4626): "*Although deep clouds show the largest variability in cloud water mass mixing ratio, the differences between CTL and other simulations are not large*".

Fig. 10: I suggest hatching insignificant areas here.
Thank you for the suggestion. We have updated the figure by masking out the regions with precipitation between -1 and 1 mm/day.

L438-440: How is rain initiated in the model? I think this is not covered in the model description?
Thanks for pointing it out. We have included additional information to describe the rain in the CASIM model. Autoconversion and accretion follows the Khairoutdinov and Kogan (2000) and the self-collection of rain droplets (with rain droplets ) and cloud droplets (with cloud droplets) is based on Beheng et al. (1994). We have added the following text to the manuscript (line 241) "*Rain formation (autoconversion and accretion) from cloud droplets follows Khairoutdinov and Kogan (2000). The self-collection of rain droplets (with rain droplets) and cloud droplets (with cloud droplets) is based on Beheng et al. (1994)*".

L474: Is this out of the total or of the ones added regionally (i.e. CTL-offREG)?
The primary aerosol contributes to 0.5% of the total particle number concentration in CTL simulation in the high-sulfur regions of the regional model. We revised the sentence to clarify it as follows (line 527) "*Overall the primary aerosol contributed to around 0.5% of the height-mean total particle concentrations in the CTL simulation below 4 km altitude in the high-sulfur regions in the regional domain (Fig. 4)*".

L479: This sentence seems a bit unmotivated. Consider revising.
Yes, it is a bit out of place. It is now moved to line 524 in the new document and we revised it to improve the flow of the text. "*The positive correlation between particle number concentrations and anthropogenic emissions in Amazonia was also found in Shrivastava et al. (2019) and Zhao et al. (2021).*"

L498: Again, if this is the reason, why does also the condensation sink decrease then?
We have added a new figure in the Appendix (Fig. A6) which shows that gas-phase $SO_2$ dominates the gas-phase content. The model grids that fall into the largest gas-phase sulfur content bin are the grids that are very close to the pollution source. Likely, these

few model grids do not have many particles or $H_2SO_4$, and thus showing low condensation sink, nucleation rates and sulfuric acid. Another figure in the Appendix (Fig. A11) shows the ratios of Aitken mode to insoluble Aitken mode aerosol number concentrations against distance from Manaus has been added. It suggests that NPF and particle growth are suppressed very close to Manaus (the pollution source) and they are enhanced farther away where pollution is moderate.

The suppression of NPF by extra primary aerosol is shown in the lower gas-phase sulfur content bins, before the significant reduction. Therefore, we specify in the manuscript that primary aerosol can only explain the suppression in nucleation rates, nucleation mode and Aitken mode aerosol for gas-phase sulfur content between $4x10^{-3}$ - $1x10^{-2}$ g m$^{-2}$ and the significant reduction in condensation sink in the highest gas-phase sulfur content bin is due to several factors mentioned in the reply above (Comment: L381-382: In addition to H2SO4 and nucleation … in page 16 of this document). To clarify the range of gas-phase sulfur content, we revised the following content at line 548 "*The nucleation mode aerosol, Aitken mode aerosol, H$_2$SO$_4$-Org nucleation rate and sulfuric acid concentration reach a plateau and subsequently have a reduction of around a factor of 4 at the gas-phase sulfur content ranging between $4x10^{-3}$ - $1x10^{-2}$ g m$^{-2}$. The reduction in nucleation rates between $4x10^{-3}$ - $1x10^{-2}$ g m$^{-2}$ is related to the increasing primary aerosol emission as it gradually becomes closer to the source of the pollution, but not where gas-phase sulfur content is very high*".

The sentence "The suppression on condensable gases, nucleation and tiny aerosols is most evident near the source of the pollution" has been replaced with (line 555) "*In our model, all the variables have significant decrease when gas-phase sulfur content is greater than $1x10^{-2}$ g m$^{-2}$ in Fig. 5 and 6. The model grids with gas-phase sulfur content greater than $1x10^{-2}$ g m$^{-2}$ contain mainly SO$_2$ and very few particles or little H$_2$SO$_4$, causing significant reductions in nucleation rates and condensation sink. Figure A11 shows the ratio of Aitken mode to insoluble Aitken mode aerosol concentrations as a function of distance from Manaus (the pollution source) at around 550 m altitude for the 6 experiments with scaled emission loadings. The ratios are low within approximately 20 km downwind of Manaus, while the ratio increases significantly at distances of 100-200 km from Manaus. It suggests suppressed NPF in very high sulfur regions and enhanced NPF in moderate sulfur regions, which is consistent to results showed in Fig. 5 and 6.*"

L510: Check this sentence, there's repetition.
We agree that it does repeat the previous content. We revised the sentence as follows (line 571) "*Reducing the aerosol concentration caused a reduction of N$_d$ by a factor of 2 in the 0.25xaero simulation compared to the CTL simulation*".

L516: Why would heterogeneous ice nucleation not being interactive weaken the "relationship" between ice and droplets? Is "interactively correlated" the right term here? I would say connected for example?
It was a typo and the sentence should state that there is no direct interaction between aerosol and ice in our model configuration, but the influence of aerosol on ice should be possible indirectly because cloud droplet concentration can influence ice in our model setup. We have deleted the sentence and revised the next one (at line 574) "*Although

*our model does not have aerosol-aware heterogeneous ice nucleation, aerosol number concentration may still influence ice indirectly through cloud droplet number concentration which can affect the number concentration of ice crystals formed via homogeneous freezing.*"

L527-528: Could you specify how a lack of prognostic supersaturation lead to a weaker effect?
The original sentence was not written in a clear way. There is a possibly that lack of prognostic supersaturation will result in different responses of cloud, instead of weakening the overall effect of anthropogenic emissions. If supersaturation is not prognostic, the model will not be able to account for the conditions in the previous timesteps and form cloud droplets independently at the current timestep. however, we do not expect We revised the sentence to clarify it (line 585): *"There is a possibility that using a more complex cloud microphysics scheme may contribute to different responses of cloud and rain. For example, the lack of prognostic supersaturation in this work may break the continuity of the evolution of the clouds and consequently the results may be different compared to with prognostic supersaturation (Fan and Khain, 2021)"*.

L528-530: I don't understand these two sentences or what the implications are for your result. In the last sentence you seem to contradict yourself?
The sentences should be clarified. In the study of Furtado and Field (2022), a surface rainfall frequency probability density function was found using the Met Office Unified Model. They found that the distribution was not affected by cloud droplet and aerosol number concentrations. The results imply that rainfall distribution is an invariant property of their model regardless of aerosol and cloud variations. We have updated the sentences as follows (line 590) "*Additionally, Furtado and Field (2022) showed a surface rainfall frequency probability function based on the Met Office Unified Model and this distribution was not altered by aerosol or cloud droplet number concentrations. Their results imply that even if aerosol may affect rainfall amount in individual model grids, rainfall distribution is an invariant property of their model*".

L531: Is this is just because your changes in emissions change the CCN concentration much much less than a factor of 4? Do you think it would be interesting to see the change in Nd plotted against the CCN concentration (or N100) in each of the simulations to see if it's completely linear?
We agree that 0.25xaero and 4xaero simulations lead to greater changes in cloud and rain properties than the rest of the simulations. This change is primarily because we directly scaled the number of aerosol particles that can be activated ($N_i$ in Equation 13, where '$i$' represents aerosol modes) in Abdul-Razaak and Ghan (2000). The scaling forced the model to generate a large number of cloud droplets, while the actual aerosol number concentration in the UKCA model remained unchanged.

It would be indeed be helpful to plot the $N_d$ against $N_{d>100nm}$ for all the simulations. We analysed the simulations and Fig.2 in this document shows the relationship between $N_d$ against $N_{d>100nm}$. The figures do not show linear relationship between aerosol particles and cloud droplets. They are not linear because of other non-linear processes such as

rainout and washout. It is not possible to interpret the $N_d$-$N_{d>100nm}$ relationship in the simulations 0.25xaero and 4xaero (although they are included here), because the changes are only applied in the cloud microphysics and the emission/formation of aerosol remain the same as CTL simulation.

[Figure]

Fig. 2 Scatter plots that show the relationship between cloud droplet number concentrations ($N_d$) and particle concentrations with diameters greater than 100 nm ($N_{D>100nm}$).

L554: Again, I am not sure correlation is the right word here.
Thanks for pointing it out. We have changed the sentence (now at line 622) as follows "*Overall, the relationships between anthropogenic emissions, aerosols, clouds and rain are complex, and the perturbations of anthropogenic emissions do not show systematic changes in cloud liquid water, cloud ice water or rain*".

L557-558: "[...] of background aerosols even without anthropogenic emissions in the regional domain and the small perturbations of Nd as a result of small changes in aerosols." The end of this sentence seems strange to me. Consider revising.
Thanks for your comment. The sentence (now at line 623) has been revised "*The insensitivity is potentially due to the environment (even when we switched off all anthropogenic emissions) already having a lot of background aerosols for cloud activation in the regional domain. Under this condition, the subsequent changes in $N_d$ are small and eventually result in insignificant changes in other cloud properties*".

---

## Author Response (AR2)

**Reply to the comments for egusphere-2025-132**

We thank the reviewer and the editor for the additional comments which further helped us improved the manuscript. We have provided replies to the comments and revisions in the manuscript. The reviewer's comments are shown below in black, and our responses are provided in blue. New sentences and paragraphs have been added to the manuscript, highlighted in green italics. Line numbers refer to those in the revised manuscript.

- L11: "However, as it becomes closer to the source..." This sentence is unclear; please consider rephrasing for clarity.

Reply: Line 11 has been revised as follows: "*However, nucleation is suppressed very close to the pollution source, resulting in lower nucleation and soluble Aitken mode aerosol number concentrations.*"

- L14: Cloud water mass and rain mass are not standard terms to my knowledge. Should it be mass mixing ratios?

Reply: Yes, they are mass mixing ratios and have been corrected in the abstract.

- L16: This may be a detail, but it seems here that a weak response is taken to imply non-linearity here. A response can be weak but linear. It is only non-linear if the response changes depending on the perturbation strength. It is not clear to me in the context what the non-linearity refers to here.

Reply: We agree that a weak response does not necessarily relate to non-linearity, and we will specify the two findings explicitly. The response of cloud and rain mass mixing ratios is weak when averaged over cloudy and or high-sulfur regions and does not scale linearly with the changes in anthropogenic emissions across the simulations with scaled anthropogenic emissions. When we altered the number of aerosols in the cloud microphysics scheme by factors of 0.25 and 4, the corresponding changes in cloud droplet number concentration was large (by around factors of 0.5 and 2). However, such significant changes in droplet number still resulted in moderate changes in cloud mass mixing ratio and precipitation. It indicates that the pathway from aerosol perturbations to rain formation is buffered by several microphysical processes and the overall response is non-linear.

The sentence at line 16 has been revised to "*If we assume our simulation has a fine enough grid resolution and an accurate representation of the relevant atmospheric processes, the simulated weak and non-linear response of cloud and rain properties to linearly scaled anthropogenic emissions suggests that the interactions among aerosol, cloud and precipitation in the Amazonian convective environment are buffered by microphysical processes.*"

- Fig. A11. Is it really so that the ratio of Aitken mode to insoluble Aitken mode is on the scale of 0 to 1e10? This seems a bit like it could be a unit error. Also, is it Aitken soluble over Aitken insoluble? Or Aitken total over Aitken insoluble? I suggest to make this clear in the caption.

R: It is soluble Aitken mode vs insoluble Aitken mode concentrations around 550 m altitude and have been corrected in the caption.

The ratio of soluble Aitken mode to insoluble Aitken mode number concentrations is unitless. The concentrations are collected around 550 m altitude where NPF is permitted, while this altitude is much higher than pollution source (at the surface). Thus, insoluble Aitken mode aerosol concentrations are low compared to soluble Aitken mode aerosol at this altitude, resulting in high ratios. We checked the distributions of the ratios and found that the majority of the ratios are between $10^2$-$10^5$. Those reaching up to $10^{10}$ are rare (see the figure below).

[Figure]

Fig. Histograms of the ratios of soluble Aitken mode to insoluble Aitken mode aerosol number concentrations at 555 m altitude.

- L264: About the description of the scaling in the 0.25xaero and 4xaero simulations: I found your response to the review comment much clearer than the current manuscript text. Consider incorporating something like: "This means we directly scale the number of particles after the activation diameter is determined, and thus do not allow the scaling to influence supersaturation, activation diameter, etc."

R: We revised the sentence to "*In this procedure, we directly scale the number of particles after the maximum supersaturation has been determined and thus do not allow the aerosol activation diameters and concentrations to be adjusted to updraft velocities or water vapour availability*." The sentence is now at line 266.

- L513-518: This sentence should be split for clarity. I also don't understand what "but we expect the absence of isoprene-NOx mechanism to be within the uncertainty of current NPF mechanism" means.

R: Yes, we agree that the sentence is better when split. We have moved the content regarding isoprene-$NO_x$ nucleation to the end of the paragraph.

We expect the effects of the absence of isoprene-$NO_x$ mechanism to be within the uncertainty of the current NPF mechanism, especially given that we have to disable upper tropospheric NPF to better match the observed particle concentrations. To better match the observations, we have investigated a range of oxidation rates of monoterpene to form HOMs to particle concentrations in the Amazon during the wet season and the dry season (Wang et al., 2023) which is to some extent will produce similar variability to the addition of a NPF mechanism based on isoprene-$NO_x$. The results indicated that enabling NPF in the upper troposphere could always produce around 1000 $cm^{-3}$ particles in the free troposphere which is much more than those observed during the GoAmazon2014/5 campaign.

We deleted the sentence to avoid confusion and added "*Our model does not include this NPF mechanism due to the absence of isoprene-$NO_x$ chemistry, but we do not expect the absence of this mechanism to significantly affect our results.*" at line 518.

- L518-520: I am not sure I understand what your conclusion is for point 2) here. Are you saying that N100 may be wrong due to upper tropospheric nucleation, or that you think this is unlikely?

R: Incorporating upper tropospheric (UT) NPF causes the model to significantly overestimate the observations. It does not necessarily mean that including UT NPF is wrong, but rather indicates a mismatch between the model and the observations. Not using UT NPF is not physically realistic because many previous studies have shown the importance of new particle formation (NPF) in the UT. Therefore, UT NPF is generally included in the models. In this study we only use the GoAmazon2014/5 observation dataset for our time period, which focused mainly on boundary layer and found very few particles in the free troposphere during the Amazonian wet season.

When we switched on UT NPF in the model, particle concentrations increased significantly in the free troposphere and the boundary layer, leading to an overestimation compared to the observations. To better match the observations, we have to suppress UTNPF in our simulations. This setup is not ideal but a compromise that likely still causes biases in the concentrations of particles greater than 100nm in diameter, and should be improved on in future simulations..

It is important to note that the aircraft measurements were primarily focused on the boundary layer and it is very uncertain how representative the observations in the free troposphere are of typical conditions in that region.

This paragraph starting from line 514 has been revised as follows:

*"(2) Upper tropospheric (UT) NPF along with subsequent downward transport, has been shown to be important for determining low-level particle concentrations (Clarke et al., 1998, 1999; Clarke and Kapustin, 2002; Merikanto et al., 2009; Wang et al., 2016; Williamson et al., 2019; Curtius et al., 2024), and it is important for Amazonia during the dry season (Andreae et al., 2018). Observations have reported bursts of particles due to NPF from organic compounds formed by isoprene with $NO_x$ (Kuhn et al., 2010; Bardakov et al., 2024; Shen et al., 2024). Our model does not include this NPF mechanism due to the absence of isoprene-$NO_x$ chemistry, but we do not expect the absence of this mechanism to significantly affect our results. In this study we focus on the wet season and only use the GoAmazon2014/5 observation dataset for our time period, which focused mainly on the boundary layer (below 2 km), although the infrequent sampling casts doubt on how representative of mean conditions these observations were. The aircraft occasionally flew between 2 - 6 km altitude and found very few particles in the free troposphere during the Amazonian wet season. Consequently, it is very uncertain how representative the observations in the free troposphere are of typical conditions in that region. When we switched on UT NPF in our model, particle concentrations increased significantly in the free troposphere and the boundary layer, leading to an overestimation compared to the observations. To better match the observations, we therefore disabled NPF above 1 km to achieve consistency between the model and observations in March 2014. This setup is not ideal but a compromise that likely still causes biases in the concentrations of particles greater than 100nm in diameter, and should be improved on in future simulations."*

- L571: I thought I had understood what you had done with the 0.25 and 4xaero simulations, but it is not clear to me why you are then getting a reduction in Nd which is half of your perturbation? Could you clarify?

R: The '$N_i$' that we perturbed in the activation scheme is not equivalent to the $N_d$ in the model output. Although the variable '$N_i$' is scaled by a factor of 0.25 and 4, the $N_d$ in the model output is influenced by several nonlinear microphysical (to aerosol number concentrations) processes every timestep (the Abdul-Razaak and Ghan (2000) droplet activation scheme itself, rain evaporation, formation of graupel, deposition, condensation etc.). Additionally, $N_d$ in Fig. 7 is derived after averaging the 3-hourly output over time and cloudy-polluted regions. As a result, the changes in $N_d$ relative to CTL simulation are not linear to the aerosol perturbations (0.25 and 4). We added the following at line 581 to clarify it:

"*The variable we perturbed in these simulations ($N_i$ in the Abdul-Razaak and Ghan (2000) droplet activation scheme, hereafter ARG2000) in the CASIM activation scheme is not directly equivalent to the model output $N_d$ since ARG2000 is non-linear between $N_i$ and $N_d$. The model output $N_d$ depends strongly on updraft speeds via the activation parameterization, and is also influenced several dynamical and microphysical processes each timestep (e.g. advection, droplet freezing, riming, or warm rain formation) and $N_d$ shown in Fig. 7 has been averaged over time and high-sulfur-cloudy regions using the 3-hourly model output. Therefore, changes in $N_d$ in these simulations do not scale directly with the aerosol perturbation relative to the CTL simulation.*"